# One-Timestep is Enough: Achieving High-performance ANN-to-SNN Conversion via Scale-and-Fire Neurons

## Abstract

Spiking Neural Networks (SNNs) are gaining attention as energy-efficient alternatives to Artificial Neural Networks (ANNs), especially in resource-constrained settings. While ANN-to-SNN conversion (ANN2SNN) achieves high accuracy without end-to-end SNN training, existing methods rely on large time steps, leading to high inference latency and computational cost. In this paper, we propose a theoretical and practical framework for single-timestep ANN2SNN. We establish the Temporal-to-Spatial Equivalence Theory, proving that multi-timestep integrate-and-fire (IF) neurons can be equivalently replaced by single-timestep multi-threshold neurons (MTN). Based on this theory, we introduce the Scale-and-Fire Neuron (SFN), which enables effective single-timestep ($T = 1$) spiking through adaptive scaling and firing. Furthermore, we develop the SFN-based Spiking Transformer (SFormer), a specialized instantiation of SFN within Transformer architectures, where spike patterns are aligned with attention distributions to mitigate the computational, energy, and hardware overhead of the multi-threshold design. Extensive experiments on image classification, object detection, and instance segmentation demonstrate that our method achieves state-of-the-art performance under single-timestep inference. Notably, we achieve 88.8% top-1 accuracy on ImageNet-1K at $T = 1$, surpassing existing conversion methods.

## 1 Introduction

Artificial Neural Networks (ANNs) have achieved significant success in a wide range of computer vision tasks, including image classification (Wu et al., 2023), object detection (Zou et al., 2023), and instance segmentation (Sharma et al., 2022). However, the high energy consumption during ANN inference has become a critical limitation. To mitigate this issue, Spiking Neural Networks (SNNs) (Gerstner & Kistler, 2002) have been proposed as a promising energy-efficient alternative. SNNs are a class of neural networks inspired by the mechanisms of biological neurons (Bohté et al., 2000; Gerstner et al., 2014). In SNNs, neurons emit spikes only when their membrane potential exceeds a predefined threshold (Mueller et al., 2021). When implemented on neuromorphic chips (Davies et al., 2018; DeBole et al., 2019; Merolla et al., 2014; Pei et al., 2019), the event-driven computation combined with the sparse and asynchronous spike-based communication enables SNNs to achieve significantly lower power consumption than ANNs (Roy et al., 2019; Yao et al., 2024). Nevertheless, the complex neuronal dynamics and the non-differentiable nature of spikes present significant challenges for training SNNs with high accuracy.

Currently, two main approaches are employed for training SNNs: direct training and ANN-to-SNN conversion (ANN2SNN). In direct training methods (Neftci et al., 2019; Zhou et al., 2023b; 2024; Zhu et al., 2022), spiking neuron states must be propagated through multiple time steps to support gradient computation. This process inherently relies on extensive backpropagation through time (BPTT), resulting in significant memory consumption and extended training time. In contrast, ANN2SNN methods (Bu et al., 2022b; Cao et al., 2015; Deng & Gu, 2021; Li et al., 2021; Rueckauer et al., 2017) replace modules in a pre-trained ANN with spiking neurons, aiming to align SNN spike firing rates with continuous activation values of the original ANN. Therefore, ANN2SNN methods do not require end-to-end SNN training, significantly reducing the memory consumption

Figure 1: Comparison between conventional multi-timestep SNNs and our single-timestep SNNs. (a) Traditional SNNs require multiple timesteps ($T > 1$) to accumulate information over time for accurate representation. (b) IF neurons emit either one spike or none, based on a single threshold $\theta$. (c) SFN use multiple thresholds to emit multiple spikes in a single timestep. (d) Our method replaces temporal accumulation with spatial threshold modulation, enabling high-accuracy inference within a single timestep ($T = 1$).

and training time requirements. Moreover, ANN2SNN methods have achieved superior inference accuracy compared to direct training methods across various visual benchmarks (Wei et al., 2024).

However, to achieve comparable performance as ANNs, current ANN2SNN approaches require large time steps (Hao et al., 2023). This reliance on large time steps inevitably increases both inference latency and computational costs, particularly in large-scale ANN conversions. This factor motivates us to achieve high-precision ANN2SNN under low latency, especially at timestep $T = 1$.

In this paper, we present the **Temporal-to-Spatial Equivalence Theory**, which provides a rigorous theoretical basis for single-timestep ANN2SNN frameworks. Specifically, we prove that multi-timestep integrate-and-fire (IF) neurons (Tal & Schwartz, 1997) are equivalent to single-timestep multi-threshold neurons (MTN). As illustrated in Figure 1, we design the **Scale-and-Fire Neuron (SFN)** based on the Temporal-to-Spatial Equivalence Theory, enabling the replacement of multi-timestep IF neurons with single-timestep SFN neurons.

SFN integrates two mechanisms: (1) a membrane-potential scaling strategy that reduces conversion error, thereby preserving accuracy at $T = 1$, and (2) an adaptive fire function with dynamic thresholding that fits the activation distribution and reduces the number of required thresholds for efficient computation. To adapt SFN to Transformers, we introduce the **SFN-based Spiking Transformer (SFormer)**, a specialized instantiation that aligns spike patterns with attention distributions.

Our main contributions are summarized as follows:

- We establish the Temporal-to-Spatial Equivalence Theory, revealing that temporal spike integration processes can be precisely reconstructed through spatial multi-threshold mechanisms within a single timestep for the first time.

- We design the Scale-and-Fire Neuron (SFN), whose core scaling mechanism combined with an adaptive fire function enables high-performance and energy-efficient ANN-to-SNN conversion under single-timestep ($T = 1$) inference.

- We propose the SFN-based Spiking Transformer (SFormer), a specialized instantiation that integrates SFN into Transformer architectures, addressing large activation variations and skewed distributions through customized fire functions and threshold configurations.

- We evaluate our method on three fundamental vision tasks: image classification, object detection, and instance segmentation. At timestep $T = 1$, our approach achieves 88.8% accuracy on ImageNet-1K, surpassing the previous state-of-the-art (SoTA) of 84.0% at $T = 2$, while reducing energy consumption by 5%, establishing a new SoTA.

## 2 RELATED WORKS

### 2.1 ARCHITECTURAL DESIGN IN ANN-TO-SNN CONVERSION

The typical ANN-to-SNN conversion inserts spiking neurons between ANN layers or substitutes ANN nonlinearities with spiking neurons to transform continuous activations into discrete spikes. Early work (Cao et al., 2015) replaced ReLU with spiking neurons, while Diehl et al. (2015) improved accuracy via weight normalization. Subsequent advances introduced soft reset neurons (Rueckauer et al., 2017; Han et al., 2020) to address spike count errors from hard resets. Recent optimizations focus on two directions: enhanced spiking neuronal parameters and models (dynamic thresholds (Sengupta et al., 2018; Zhang et al., 2024), optimized initialization (Bu et al., 2022a), burst-spike (Li & Zeng, 2022) and signed neurons (Wang et al., 2022), and negative spikes (Li et al., 2022a)) and ANN adaptations (activation quantization (Esser et al., 2020), trainable clipping layers (Ho & Chang, 2021), and ReLU alternatives (Bu et al., 2022b; Ding et al., 2021; Han et al., 2023; Jiang et al., 2023; Wang et al., 2023)). For Transformer-SNN conversion, Jiang et al. (2024) proposed Universal Group Operators and a Temporal-Corrective Self-Attention Layer, but faced accuracy gaps with the ANN. Huang et al. (2024) introduced Expectation Compensation and Multi-Threshold Neurons to mitigate the performance gap, but their methods remain ineffective at $T = 1$.

Existing methods rely on multi-timestep accumulation. As $T$ decreases, the approximation error grows, although latency and energy consumption are reduced. Our method directly enables high-performance ANN-to-SNN conversion at $T = 1$.

### 2.2 ERROR ANALYSIS IN ANN-TO-SNN CONVERSION

The performance degradation in ANN2SNN originates from the inherent discrepancy between continuous activations and discrete spikes. Previous studies have categorized these conversion errors into three types: clipping error, quantization error, and unevenness error (Bu et al., 2022b). For Transformer architectures, an additional non-linearity error emerges (Huang et al., 2024). Existing ANN2SNN approaches have attempted to mitigate these errors through various strategies: optimizing activation functions to reduce clipping and quantization errors (Bu et al., 2022b; Yan et al., 2021), developing residual membrane potential compensation for unevenness error (Hao et al., 2023), and proposing Expectation Compensation to address non-linear error (Huang et al., 2024).

In our single-timestep ANN2SNN framework, the unevenness errors and the non-linear errors are inherently eliminated since no temporal integration is required. Consequently, our methodology focuses exclusively on minimizing the remaining clipping and quantization errors.

## 3 METHODOLOGY

In this section, we first establish the Temporal-to-Spatial Equivalence Theory as the theoretical foundation for **single-timestep** ANN2SNN frameworks. Then, we propose the Scale-and-Fire Neuron (SFN), which enables **high-accuracy** spiking computation under single-timestep. The overall framework of our ANN2SNN method is illustrated in Figure 2. Finally, we design the SFN-based Spiking Transformer (SFormer), a specialized instantiation that converts Transformer architectures into SNNs by matching activation distributions and adapting to the self-attention mechanism.

### 3.1 TEMPORAL-TO-SPATIAL EQUIVALENCE THEORY

#### 3.1.1 INTEGRATE-AND-FIRE (IF) NEURON

IF neuron is among the most popular neurons in SNNs, due to the balance between bio-plausibility and computing efficiency. The dynamics with a soft reset mechanism is formulated as:

$$h(t) = v(t - 1) + x(t), \tag{1}$$

$$o(t) = \theta * \mathcal{H}(h(t) - \theta), \tag{2}$$

$$v(t) = h(t) - o(t), \tag{3}$$

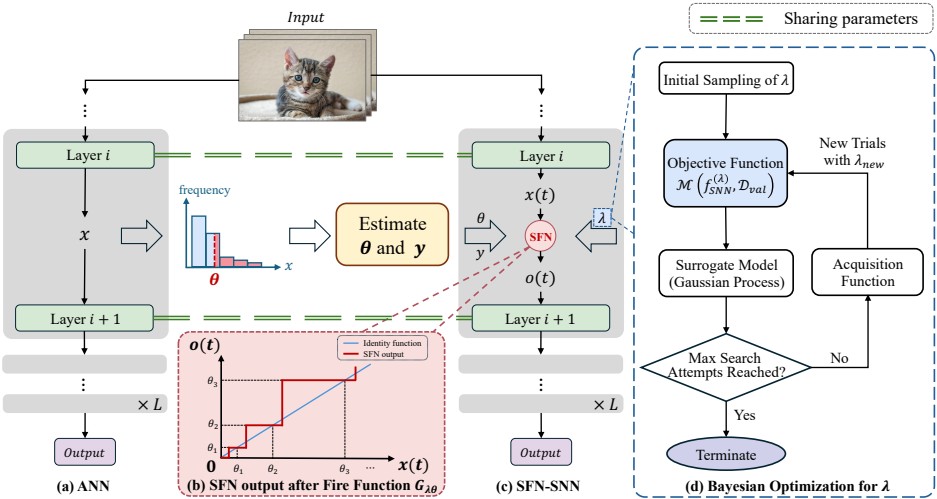

Figure 2: The framework of our ANN2SNN method. (a) The standard ANN performs a forward pass while collecting activation values from each layer for parameter estimation. (b) The output curve of the SFN. (c) The SFN enables single-timestep inference by adopting parameters estimated from the ANN activation distribution, and further refined via Bayesian optimization. (d) Bayesian Optimization for efficiently searching for the optimal scaling factor $\lambda$, where the objective function evaluates the performance of the SNN configured with $\lambda$ on a validation dataset.

where $t \in [1, T]$ represents the discrete timestep. At each timestep $t$, $v(t)$ and $h(t)$ are the membrane and latent membrane potentials, $x(t)$ is the input, and $o(t)$ is the output. $\mathcal{H}(\cdot)$ denotes the standard Heaviside step function, which outputs 1 if the input is non-negative, and 0 otherwise.

The threshold $\theta$, which is predefined for each neuron, controls spike generation through the Heaviside step function. According to Rueckauer et al. (2016) and Diehl et al. (2015), setting $\theta$ close to the maximum activation at each location is an effective means of reducing conversion errors.

The output over multiple timesteps $T$ is computed as the average firing rate $\bar{o}$, formulated as:

$$\bar{o} = \frac{1}{T} \sum_{t=1}^{T} o(t). \tag{4}$$

Due to the residual membrane potential $v(T)$, a discrepancy arises between the average input $\bar{x} = \frac{1}{T} \sum_{t=1}^{T} x(t)$ and the average firing rate $\bar{o}$, resulting in conversion errors:

$$\epsilon = |\bar{x} - \bar{o}| = \frac{|v(T) - v(0)|}{T}. \tag{5}$$

### 3.1.2 MULTI-THRESHOLD NEURON (MTN)

To emulate the multi-timestep behavior of IF neurons within a single timestep, we propose the MTN that supports multiple spikes per timestep, with the spike count determined by successive threshold-crossing events. The dynamics of the MTN can be reformulated by rewriting equation 2 as:

$$o(t) = \theta_M * \text{clip}\left(\left\lfloor \frac{1}{\theta_M} h(t) \right\rfloor, 0, N\right), \tag{6}$$

where $\theta_M$ denotes the base threshold, $\lfloor \cdot \rfloor$ denotes the floor function, $\text{clip}(\cdot, v_{\min}, v_{\max})$ represents the value clipping operation between lower bound $v_{\min}$ and upper bound $v_{\max}$, and $N$ is the maximum number of spikes that can be emitted within a single timestep.

### 3.1.3 EQUIVALENCE THEOREMS

By analyzing the input-output behavior of IF and MTN, we propose the following Temporal-to-Spatial Equivalence Theory:

Table 1: Time and energy complexity ($T$: IF neuron timesteps, $N$: number of multi-threshold).

|  | Neuron Update Time | Non-neuronal Time | Average Firing Rate |
|---|---|---|---|
| IF (single timestep) | $O(1)$ | $O(1)$ | $O(1)$ |
| IF (total $T$ timesteps) | $O(T)$ | $O(T)$ | $O(T)$ |
| MTN (single timestep, $N$ thresholds) | $O(N)$ | $O(1)$ | $O(N)$ |

**Theorem 1.** *Under the condition that both the input and the initial membrane potential of the IF neuron are non-negative and bounded by $\theta$, the response of an appropriately constructed MTN at single-timestep is equivalent to that of an IF neuron integrated over $T$ timesteps.*

**Theorem 2.** *Under the conditions of Theorem 1, and given identical network structure and parameters, an SNN equipped with multi-timestep IF neurons is functionally equivalent to one equipped with single-timestep MTNs if all operations other than the neuronal operations are linear.*

Theorem 1 establishes the equivalence between temporal spike integration and spatial multi-threshold mechanisms under ideal conditions. Building on Theorem 1, Theorem 2 extends this equivalence to the network level. In practice, signed and over-threshold inputs violate the ideal assumptions of Theorem 1. To handle arbitrary inputs, we adopt dual-IF and dual-MTN (see Appendix A) to separately process positive and negative components of the input.

**Theorem 3.** *The single-timestep dual-MTN closely approximates the behavior of the multi-timestep dual-IF, and their discrepancy is bounded. Specifically, the output error satisfies*

$$|\bar{o}_{IF} - o_M| \leq \frac{C_v + C_\theta}{T}, \tag{7}$$

where $\bar{o}_{IF}$ denotes the average firing rate of the dual-IF over $T$ timesteps, $o_M$ is the single-timestep dual-MTN output, and the constant $C_v$ and $C_\theta$ depends only on residual membrane potentials and the dual thresholds, respectively. Consistent with the conversion error control in equation 5, $C_v$ and $C_\theta$ admit a reasonably small upper bound.

Theorem 3 establishes approximate equivalence in practice. Detailed proofs are given in Appendix A, and experimental verification is shown in Appendix D.

### 3.1.4 DISCUSSION

**Time and energy analysis.** We decompose the time cost into *neuron update time* and *non-neuronal time* (e.g., linear operation), where the latter typically dominates. The energy cost is primarily driven by the average firing rate. As summarized in Table 1, when $N \approx T$, a single-timestep MTN executes the non-neuronal operation only once and thus achieves an almost $T\times$ speedup over the IF neuron, while their firing rate is theoretically comparable.

**Limitations and improvements.** In large models, achieving high performance with IF neurons may require very large $T$; a naive conversion to MTNs yields a very large $N$ and a very small $\theta$, which increases energy and is hardware-unfriendly. Our key observation is that single-timestep performance stems from two mechanisms, namely membrane-potential scaling and an adaptive fire function. This observation motivates us to design a new single-timestep neuron that realizes these mechanisms directly, rather than adhering to the previous multi-threshold structure.

### 3.2 SCALE-AND-FIRE NEURON (SFN)

Based on the preceding theory, we introduce the Scale-and-Fire Neuron (SFN). Inserting SFN between adjacent ANN layers enables single-timestep ($T = 1$) ANN2SNN.

### 3.2.1 MODELING THE DYNAMICS OF SFN

The dynamics of SFN can be reformulated by rewriting Eq. equation 6 as:

$$o(t) = \lambda\theta * G_{\lambda\theta}(h(t)), \tag{8}$$

where the fire function $G_{\lambda\theta} : \mathbb{R} \to \mathbb{Z}$ is a piecewise constant function, $\theta$ denotes the threshold of the IF neuron and $\lambda \in (0, 1]$ is a scaling factor that adjusts the firing sensitivity of the neuron.

In practice, we set $\theta$ to the top-$p\%$ activation within each ANN layer (with $p$ as a hyperparameter). This percentile thresholding mitigates sensitivity to outliers and distributional skew and places $\theta$ above the majority of instantaneous inputs in most cases.

The subsequent sections detail the derivation of the scaling factor $\lambda$ and the fire function $G_{\lambda\theta}$.

### 3.2.2 BAYESIAN OPTIMIZATION FOR SCALING FACTOR

The scaling factor $\lambda$ of SFN requires task- and architecture- specific design. By adjusting $\lambda$ as a continuous variable, the threshold $\lambda\theta$ is modulated accordingly, which in turn controls the granularity of spike generation. This adjustable design allows SFN to flexibly adapt to diverse tasks and achieves an effective balance between conversion precision and spike sparsity.

We adopt Bayesian Optimization to efficiently determine the optimal value of $\lambda$. The optimization objective is defined as follows:

$$\lambda^* = \arg \max_{\lambda \in (0,1]} \mathcal{M}\left(f_{\text{SFN}}^{(\lambda)}, \mathcal{D}_{\text{val}}\right),\tag{9}$$

where $\mathcal{M}(\cdot, \cdot)$ denotes the evaluation metric used to measure model accuracy (e.g., top-1 accuracy), $f_{\text{SFN}}^{(\lambda)}$ represents the SFN model constructed with a given $\lambda$, and $\mathcal{D}_{\text{val}}$ is the validation dataset used for optimization. The optimization process is illustrated in Figure 2d.

### 3.2.3 FORMULATION OF THE FIRE FUNCTION

The optimization objective of SFN is essentially to minimize the conversion errors. Given different models, we determines the fire function $G_{\lambda\theta}(\cdot)$ based on the distribution of pre-activation values.

The fire function $G_{\lambda\theta}(\cdot)$ is required to be monotonically non-decreasing and to take only a finite number of discrete values. Specifically, we assume its discontinuities occur at ordered thresholds $\theta_1 < \ldots < \theta_N$, with the corresponding function values $y_1 < \ldots < y_N$, where $y_i \in \mathbb{Z}^+$. Under these assumptions, we can derive the explicit form of $G_{\lambda\theta}(\cdot)$:

$$G_{\lambda\theta}(h(t)) = \begin{cases} 0, & 0 \le h(t) < \theta_1 \\ y_i, & \theta_i \le h(t) < \theta_{i+1} \\ y_N, & \theta_N \le h(t). \end{cases}\tag{10}$$

To make the SFN approximate the identity mapping, we set the thresholds to be proportional to the discrete output levels, i.e., $\theta_i = \lambda\theta y_i$. Following Bu et al. (2022a), the initial membrane potential is set to $v(0) = \frac{\lambda\theta}{2}$. Under single-timestep, the SFN input–output relation admits:

$$o = \begin{cases} 0, & 0 \le x < \lambda\theta\left(y_1 - \frac{1}{2}\right) \\ \lambda\theta y_i, & \lambda\theta\left(y_i - \frac{1}{2}\right) \le x < \lambda\theta\left(y_{i+1} - \frac{1}{2}\right) \\ \lambda\theta y_N, & \lambda\theta\left(y_N - \frac{1}{2}\right) \le x. \end{cases}\tag{11}$$

In equation 11, the first two cases describe in-range quantization, where the output $o$ is snapped to the nearest level $\lambda\theta y_i$; the deviation $|o - x|$ constitutes the *quantization error*. The last case corresponds to saturation when $x$ exceeds the admissible range, yielding the *clipping error*. To reduce both errors, fit the activation distribution, and avoid redundant thresholds, SFN selects thresholds by *activation density*: denser regions are assigned more, tighter intervals, whereas sparse regions use fewer, wider intervals. The outermost thresholds are placed near the distribution tails, lowering the probability of saturation and thus clipping. Complementarily, the *scaling* mechanism adjusts the step size $\Delta = \lambda\theta$, which reduces the dominant quantization error.

### 3.3 SFN-BASED SPIKING TRANSFORMER (SFORMER)

To exploit the performance and scalability of Transformer architectures in ANN2SNN conversion, we adopt Transformer as the base ANN model. However, the activations of Transformer exhibit large variation in magnitude across layers, and activations after the SoftMax in self-attention are highly non-uniform. To address these challenges, we propose SFN-based Spiking Transformer (SFormer), an ANN2SNN framework built upon SFN and tailored to Transformer architectures.

### 3.3.1 DESIGN OF THE FIRE FUNCTION IN SFORMER

To adapt to the highly skewed activation distribution in Transformer, the fire function $G_{\lambda\theta}(\cdot)$ takes $2M$ nonzero values:

$$y_k = \begin{cases} k, & 1 \le k \le M \\ M - 1 + 2^{k-M}, & N + 1 \le k \le 2M. \end{cases} \quad (12)$$

Following prior spike-transformer designs that use separate positive and negative thresholds (Huang et al., 2024), we independently assign thresholds for positive and negative activations, for a total of $N = 4M$ thresholds. The thresholds $\{\theta_k^+\}_{k=1}^{2M}$ and $\{\theta_k^-\}_{k=1}^{2M}$ in $G_{\lambda\theta}(\cdot)$ are defined as:

$$\theta_k^+ = \lambda\theta^+ y_k, \quad 1 \le k \le 2M \quad (13)$$

$$\theta_k^- = -\lambda\theta^- y_k, \quad 1 \le k \le 2M \quad (14)$$

where $\theta^+$ and $-\theta^-$ denote as positive and negative base threshold, which correspond to the top-$p\%$ largest positive and negative activation values in each layer, respectively.

### 3.3.2 THRESHOLD ADAPTATION FOR SOFTMAX IN SELF-ATTENTION

The activation distribution after the SoftMax operation in the self-attention mechanism of Transformer exhibits a highly non-uniform pattern, where a few extremely large values—far beyond typical activations—play a decisive role in the prediction of the model. To address this, we explicitly estimate the maximum activation at each SoftMax layer:

$$o^{\text{softmax}} = \max_{o \in \text{SoftMax Output}} o. \quad (15)$$

Then we use the maximum value to configure the upper bound of the fire function response, by setting the threshold as:

$$\theta^{\text{softmax}} = \frac{o^{\text{softmax}}}{\max_{1 \le k \le 2M} |y_k|} = \frac{o^{\text{softmax}}}{M - 1 + 2^M}. \quad (16)$$

Our strategy ensures the high-activation regions are not clipped, while preventing the post-SoftMax activation distribution from degenerating into a binary 0–1 pattern in our single-timestep SNN.

## 4 EXPERIMENTS

### 4.1 DATASETS AND IMPLEMENTATION DETAILS

**Datasets** We extensively evaluate our method on three tasks: image classification, object detection, and instance segmentation. (i) For image classification, we use the ImageNet-1K (Deng et al., 2009) dataset, a large-scale benchmark for visual recognition with 1,000 object categories. We conduct all evaluations on its standard validation set. (ii) For object detection and instance segmentation, we adopt the COCO-2017 (Lin et al., 2014) dataset, a challenging benchmark featuring complex real-world scenes with bounding box and mask annotations for 80 categories. We perform evaluation exclusively on the validation split.

**Implementation Details** For image classification, we convert pre-trained Vision Transformers: ViT-Base/16 and ViT-Large/16 (Vaswani et al., 2017) (224×224 input), and EVA (Fang et al., 2023) (336×336 input). For object detection and instance segmentation, we integrate EVA (1536×1536 input) with Detectron2 (Wu et al., 2019). All SFNs use $N = 4M$ thresholds defined by equation 13 and equation 14 where $M = 8$. Thresholds $\theta^+$, $-\theta^-$ are set at percentile $p = 1$, except for post-SoftMax SFNs which employ equation 16. More details are provided in Appendix E.1.

### 4.2 COMPARISON TO STATE-OF-THE-ART

**Performance on ImageNet-1K** For image classification, Table 2 presents the performance comparison with existing SNN methods on the ImageNet-1K dataset. Our method (SFormer) achieves the best accuracy of 88.8% at $T=1$ under the EVA architecture, significantly surpassing previous

Table 2: Comparison with state-of-the-art methods on ImageNet-1K image classification. The best performance is marked in **bold**.

| Type | Method | Arch. | Param. (M) | T | Accuracy (%) |
|------|--------|-------|-----------|---|-------------|
| directly training | Spikingformer (Zhou et al., 2023a) | Spikingformer-4-384-400E | 66.34 | 4 | 75.8 |
| | Spike-driven Transformer (Yao et al., 2023) | Spiking Transformer-8-768 | 66.34 | 4 | 77.0 |
| | Spikeformer (Li et al., 2022c) | Spikeformer-7L/3×2×4 | 38.75 | 4 | 78.3 |
| | SNN-ViT (Wang et al., 2025) | SNN-ViT-8-512 | 53.7 | 4 | 80.2 |
| ANN-to-SNN | SRP (Hao et al., 2023) | VGG-16 | 138 | 4 | 66.5 |
| | WFWC (Yang et al., 2025) | VGG-16 | 138 | 46 | 73.2 |
| | QCFS (Bu et al., 2022b) | VGG-16 | 138 | 74 | 73.3 |
| | OPI (Bu et al., 2022a) | VGG-16 | 138 | 128 | 74.2 |
| | QFFS (Li et al., 2022a) | VGG-16 | 138 | 8 | 73.1 |
| | ECMT (Huang et al., 2024) | ViT-Base/16 | 86 | 4 | 69.9 |
| | | ViT-Large/16 | 307 | 2 | 75.3 |
| | | EVA | 1074 | 2 | 84.0 |
| | SFormer (ours) | ViT-Base/16 | 86 | 1 | 70.7 |
| | | ViT-Large/16 | 307 | 1 | 82.4 |
| | | EVA | 1074 | 1 | **88.8** |

Table 3: Comparison with state-of-the-art methods on COCO-2017 object detection. the best performance is marked in **bold**.

| Type | Method | Arch. | Param. (M) | T | mAP@0.5:0.95 | mAP@0.5 |
|------|--------|-------|-----------|---|--------------|---------|
| directly training | SpikeYOLO (Luo et al., 2024) | SpikeYOLO | 68.8 | 1 | 48.9 | 66.2 |
| | EMS-YOLO (Su et al., 2023) | EMS-ResNet34 | 26.9 | 4 | 30.1 | 50.1 |
| | MSD (Li et al., 2025) | MSD | 7.8 | – | 45.3 | 62.0 |
| | SpikeDet (Fan et al., 2025) | SpikeDet-Small | 22 | 1 | 46.2 | 62.6 |
| | | SpikeDet-Medium | 48.2 | 1 | 48.0 | 64.8 |
| | | SpikeDet-Large | 75.2 | 1 | 49.6 | 66.5 |
| ANN-to-SNN | Spiking-YOLO (Kim et al., 2020) | Tiny-YOLO | 10.2 | 3500 | – | 25.7 |
| | bayesian optimization (Kim et al., 2021) | Tiny-YOLO | 10.2 | 500 | – | 21.1 |
| | spike calibration (Li et al., 2022b) | VGG-16 | 138 | 512 | – | 45.1 |
| | | ResNet50 | 25.6 | 512 | – | 45.4 |
| | BSNNs (Muhammad Yasir & Kim, 2025) | ResNet34 | 32 | – | – | 47.6 |
| | SUHD (Qu et al., 2025) | YOLOv5s | 7.2 | 4 | – | 54.6 |
| | SFormer (ours) | YOLOv5s | 7.2 | 1 | 34.8 | 54.9 |
| | | EVA | 1072 | 1 | **60.3** | **78.2** |

ANN-to-SNN approaches such as ECMT (84.0%) using the same backbone. In addition, SFormer is scalable to larger models with significantly more parameters, such as ViT-Large and EVA. Notably, among all directly trained and conversion-based SNNs, SFormer achieves the highest accuracy with the lowest latency, highlighting its superior efficiency and scalability. These results demonstrate the effectiveness of our method in enabling high-performance SNN inference with minimal timesteps.

**Performance on COCO-2017** For object detection, Table 3 compares our method (SFormer) with representative directly trained and ANN-to-SNN conversion-based spiking object detectors on the COCO-2017 dataset. Among all methods, SFormer achieves the highest mAP@0.5 score of 78.2%, significantly outperforming both directly trained SNNs and conversion-based baselines. Notably, SFormer operates at timestep $T = 1$, while many prior methods rely on hundreds or even thousands of timesteps to accumulate sufficient accuracy. These results highlight the ability of SFormer to deliver high-accuracy detection under ultra-low-latency constraints.

For instance segmentation, our SFormer achieves a strong performance with 50.7% mAP@0.5:0.95 and 75.1% mAP@0.5 on the COCO-2017 dataset using the EVA backbone at timestep $T = 1$. This result further confirms the effectiveness of SFormer in various vision tasks.

### 4.3 EVALUATION OF ENERGY CONSUMPTION

Table 4 reports the accuracy and energy ratio (see Appendix E.3) of our method (SFormer) under different architectures on the ImageNet-1K dataset. Across all backbones, SFormer maintains competitive accuracy at a single timestep ($T = 1$) while achieving substantial energy reduction. For

Table 4: Accuracy and energy consumption ratio on ImageNet-1K dataset.

| Arch. | Accuracy/Energy | Original (ANN) | ECMT (Huang et al., 2024) | | | SFormer (Ours) | |
|-------|-----------------|----------------|---------|---------|---------|---------|---------|
| | | | $T=1$ | $T=2$ | $T=4$ | $T=1$ | $T=2$ |
| ViT-Base/16 | Acc. (%) | 80.8 | 0.2 | 20.8 | 70.0 | 70.6 | 77.6 |
| | Energy Ratio | 1.00 | 0.06 | 0.19 | 0.51 | 0.17 | 0.46 |
| ViT-Large/16 | Acc. (%) | 84.9 | 3.6 | 75.3 | 83.2 | 82.4 | 84.2 |
| | Energy Ratio | 1.00 | 0.05 | 0.17 | 0.42 | 0.15 | 0.36 |
| EVA | Acc. (%) | 89.6 | 2.4 | 84.0 | 88.6 | 88.8 | 89.4 |
| | Energy Ratio | 1.00 | 0.06 | 0.20 | 0.49 | 0.19 | 0.47 |

Table 5: Ablation study on ImageNet-1K with the EVA architecture at $T=1$. The best performance is marked as **bold**.

| Method | Scaling Strategy | Fire Function | Accuracy |
|--------|------------------|---------------|----------|
| Baseline | ✗ | Linear | 4.4 |
| - | ✗ | Exponential | 1.6 |
| - | ✗ | SFN | 4.9 |
| - | ✓ | Linear | 84.8 |
| - | ✓ | Exponential | 65.7 |
| Ours | ✓ | SFN | **88.8** |

instance, under the EVA architecture, SFormer incurs only a 0.8% accuracy drop compared to the original ANN, with an 81% decrease in energy consumption. In contrast to ECMT, which suffers from severe accuracy degradation under low latency, SFormer delivers significantly higher accuracy with comparable or even lower energy costs, underscoring the effectiveness of SFormer for energy-efficient, low-latency inference.

### 4.4 ABLATION STUDIES

Table 5 presents the ablation results on ImageNet-1K under $T=1$, evaluating the impact of the firing function design and threshold scaling strategy. Without applying the scaling strategy, all firing functions yield poor accuracy (below 5%), primarily due to insufficient spike activity. This observation aligns with the performance degradation seen in existing MTN methods, which typically suffer from severe accuracy loss under $T=1$, as they do not incorporate a scaling factor. When the scaling strategy is applied, the accuracy improves significantly across all firing functions, with the linear and exponential variants achieving 84.8% and 65.7%, respectively. Our full method, which combines the scaling strategy with the proposed SFN firing function, achieves the highest accuracy of 88.8%. This result highlights the effectiveness of both components and underscores the importance of the scaling factor in achieving accurate single-timestep ANN2SNN, setting our method apart from other MTN designs that experience degradation in this scenario.

We further analyze the impact of the scaling factor $\lambda$, the normalization scale $p$ and the number of total thresholds $N$ on model performance and energy efficiency. Detailed ablation results are provided in Appendix B.

### 5 CONCLUSIONS

In this paper, we propose an ANN2SNN framework that combines the high accuracy of ANN-Transformer and low power of SNN, **with only a single timestep**. We establish the Temporal-to-Spatial Equivalence Theory as a rigorous basis single-timestep SNN computation and design the Scale-and-Fire Neuron (SFN) with membrane-potential scaling and adaptive firing. Our analysis highlights the decisive roles of the scaling factor and firing strategies, providing practical guidance for high-performance single-timestep conversion. Although tailored to single-timestep, SFN can naturally extend to multi-timestep settings. We instantiate SFN in Transformers as SFormer and validate its effectiveness across models up to 1B parameters and diverse vision tasks, demonstrating strong efficiency and scalability. The high accuracy (ImageNet 88.8%) combined with low energy of our methods demonstrate the practical potential for more real-world applications. A current limitation is that we have not yet extended our approach to language models; scaling to spiking LLMs remains future work.

## 6 ETHICS STATEMENT

We affirm compliance with the ICLR Code of Ethics. Our study uses only publicly available datasets (ImageNet-1K, COCO-2017, CIFAR-10) under their respective licenses and standard splits; no human-subject experiments, personally identifiable information, or sensitive attributes are involved. Code and models will be released to support transparent verification. We are unaware of conflicts of interest or legal/privacy concerns. A brief note on LLM usage for language polishing is provided in Appendix F.

## 7 REPRODUCIBILITY STATEMENT

We have taken several steps to facilitate reproducibility. Theoretical assumptions and complete proofs are provided in Appendix A. Implementation details needed to reproduce models, thresholds, and inference settings are described in Sec. 4.1 and Appendix E.1. The energy metric is specified in Appendix D. We will submit anonymized source code as supplementary material, including the SFN implementation, evaluation codes and inference scripts.

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

## A  Proofs of the Temporal-to-Spatial Equivalence Theory

This appendix provides the formal mathematical proofs of our three main theorems in the Temporal-to-Spatial Equivalence Theory.

**Theorem 1.** *Under the condition that both the input and the initial membrane potential of the IF neuron are non-negative and bounded by $\theta$, the response of an appropriately constructed MTN at single-timestep is equivalent to that of an IF neuron integrated over $T$ timesteps.*

***proof***: Consider an IF neuron receiving inputs $\{x_{IF}(t)\}_{1\leq t\leq T}$ and a MTN with initial input $x_M$ satisfying:

$$x_M = \frac{1}{T}\sum_{t=1}^{T} x_{IF}(t). \tag{17}$$

We demonstrate the equivalence by showing the average of the IF outputs $\{o_{IF}(t)\}_{1\leq t\leq T}$ is equal to the MTN output $o_M$ at initial timestep. The IF dynamics over $T$ timesteps are:

$$h(t) = v(t-1) + x_{IF}(t), \tag{18}$$

$$o_{IF}(t) = \theta * \mathcal{H}(h(t) - \theta), \tag{19}$$

$$v(t) = h(t) - o_{IF}(t). \tag{20}$$

From equation 18, equation 19 with equation 20, we derive the membrane potential relation:

$$o_{IF}(t) = x_{IF}(t) + v(t-1) - v(t). \tag{21}$$

Summing over T timesteps yields:

$$\sum_{t=1}^{T} o_{IF}(t) = \sum_{t=1}^{T} x_{IF}(t) + v(0) - v(T). \tag{22}$$

$$\frac{1}{T}\sum_{t=1}^{T} o_{IF}(t) = \frac{\theta}{T}\left(\frac{1}{\theta}\sum_{t=1}^{T} o_{IF}(t)\right) = \frac{\theta}{T}\left(\frac{1}{\theta}\sum_{t=1}^{T} x_{IF}(t) + \frac{v(0)}{\theta} - \frac{v(T)}{\theta}\right) \tag{23}$$

Under the condition that the input is non-negative and does not exceed $\theta$, we can show $0 \leq v(t) < \theta$ by induction on $t$. Thus:

$$0 \leq \frac{v(T)}{\theta} < 1. \tag{24}$$

Since $\mathcal{H}(h(t) - \theta)$ must be integer-valued, equation 19, equation 23 and equation 24 imply:

$$\frac{1}{T}\sum_{t=1}^{T} o_{IF}(t) = \frac{\theta}{T}\left\lfloor\frac{1}{\theta}\sum_{t=1}^{T} x_{IF}(t) + \frac{v(0)}{\theta}\right\rfloor. \tag{25}$$

Now consider the MTN with $N \geq T+1$, threshold $\theta_M = \frac{\theta}{T}$, and initial potential $v_M(0) = \frac{v(0)}{T}$. Its output $o_M$ is given by:

$$o_M = \theta_M * \text{clip}\left(\left\lfloor\frac{1}{\theta_M}(x_M + v_M(0))\right\rfloor, 0, N\right) = \frac{\theta}{T}\text{clip}\left(\left\lfloor\frac{T}{\theta}(x_M + \frac{v(0)}{T})\right\rfloor, 0, N\right). \tag{26}$$

Since $x \leq \theta$ and $v(0) \leq \theta$, equation 26 simplifies to

$$o_M = \frac{\theta}{T}\left\lfloor\frac{T}{\theta}x_M + \frac{v(0)}{\theta}\right\rfloor. \tag{27}$$

Combining equation 17, equation 25, and equation 27, we arrive at the conclusion of equivalence.

**Theorem 2.** *Under the conditions of Theorem 1, and given identical network structure and parameters, an SNN equipped with multi-timestep IF neurons is functionally equivalent to one equipped with single-timestep MTNs if all operations other than the neuronal operations are linear.*

***proof***: Let Model I be the SNN with IF neurons operating over $T$ timesteps and Model II be the SNN with single-timestep MTNs, both comprising $L$ layers with identical architecture and weights. Denote the layer mapping operation of layer $l$ as $\phi^l$ for both models.

For Model I (Multi-timestep IF neuron): At timestep $t \in \{1, \ldots, T\}$, the neuron before layer $l$ receives input $x_{IF}^l(t)$ and emits output $o_{IF}^l(t)$, with the inference process governed by:

$$o_{IF}^l(t) = f_{IF}\left(x_{IF}^l(t), \text{state}\right), \tag{28}$$

$$x_{IF}^{l+1}(t) = \phi^l\left(o_{IF}^l(t)\right). \tag{29}$$

For Model II (Single-timestep MTN): The neuron before layer $l$ processes input $x_M^l$ and emits output $o_M^l$ in a single timestep:

$$o_M^l = f_M\left(x_M^l\right), \tag{30}$$

$$x_M^{l+1} = \phi^l\left(o_M^l\right), \tag{31}$$

where $f_{IF}$ and $f_M$ represent the neuronal dynamics of their respective models.

Assume that the average input $\bar{x}_{IF}^l$ before layer $l$ in Model I is equal to $x_M^l$. Under the conditions specified in Theorem 1, the two neurons before layer $l$ satisfy the equivalence relation:

$$o_M^l = \frac{1}{T}\sum_{t=1}^T o_{IF}^l(t). \tag{32}$$

By the linearity of $\phi^l$, we can derive that:

$$\begin{aligned}
\bar{x}_{IF}^{l+1} &= \frac{1}{T}\sum_{t=1}^T x_{IF}^{l+1}(t) = \frac{1}{T}\sum_{t=1}^T \phi^l\left(o_{IF}^l(t)\right) \\
&= \phi^l\left(\frac{1}{T}\sum_{t=1}^T o_{IF}^l(t)\right) = \phi^l\left(o_M^l\right) = x_M^{l+1}.
\end{aligned} \tag{33}$$

In particular, when $l = 1$, $\bar{x}^1 = x^1$. The equivalence can be derived by induction on layer index $l$.

**Theorem 3.** *The single-timestep dual-MTN closely approximates the behavior of the multi-timestep dual-IF, and their discrepancy is bounded. Specifically, the output error satisfies*

$$|\bar{o}_{IF} - o_M| \leq \frac{C_v + C_\theta}{T}, \tag{34}$$

*where $\bar{o}_{IF}$ denotes the average firing rate of the dual-IF over $T$ timesteps, $o_M$ is the single-timestep dual-MTN output, and the constant $C_v$ and $C_\theta$ depends only on residual membrane potentials and the dual thresholds, respectively. Consistent with the conversion error control in equation 5, $C_v$ and $C_\theta$ admit a reasonably small upper bound.*

***proof***: We first formalize the dynamics of the dual-IF over $T$ timesteps as follows:

$$h^\pm(t) = v^\pm(t-1) + x_{IF}^\pm(t), \tag{35}$$

$$o_{IF}^\pm(t) = \theta^\pm \, \mathcal{H}\big(h^\pm(t) - \theta^\pm\big), \tag{36}$$

$$v^\pm(t) = h^\pm(t) - o_{IF}^\pm(t), \tag{37}$$

where $t = 1, \ldots, T$, $x_{IF}^+(t) = \max\{0, x_{IF}(t)\}$ and $x_{IF}^-(t) = -\min\{0, x_{IF}(t)\}$ denote the positive/negative parts of the input, and $\theta^+, \theta^- > 0$ are the thresholds for the positive and negative branches, respectively. The signed input and output are $x_{IF}(t) = x_{IF}^+(t) - x_{IF}^-(t)$ and $o_{IF}(t) = o_{IF}^+(t) - o_{IF}^-(t)$. Here $\mathcal{H}(\cdot)$ is the Heaviside step function.

In contrast, the single-timestep input–output relation of the dual-MTN can be written as

$$o_M = \begin{cases} \theta_M^+ \,\mathrm{clip}\left(\left\lfloor \dfrac{x_M + v_M^+(0)}{\theta_M^+}\right\rfloor, 0, N\right), & x_M \geq 0, \\[3mm] -\theta_M^- \,\mathrm{clip}\left(\left\lfloor \dfrac{-x_M + v_M^-(0)}{\theta_M^-}\right\rfloor, 0, N\right), & x_M < 0, \end{cases} \tag{38}$$

where $x_M$ (cf. equation 17) is the input to the dual-MTN, $\theta_M^+, \theta_M^- > 0$ are the positive/negative thresholds, $v_M^\pm(0) \in [0, \theta_M^\pm)$ are the initial residual potentials.

Without loss of generality, assume $x_M \geq 0$ and consider a dual-MTN with $N \geq T + 1$ sufficiently large, threshold $\theta_M^\pm = \frac{\theta^\pm}{T}$, and initial potential $v_M^\pm(0) = \frac{v^\pm(0)}{T}$. Then

$$o_M = \frac{\theta^\pm}{T} \left\lfloor \frac{Tx_M}{\theta^+} + \frac{v^+(0)}{\theta^+} \right\rfloor. \tag{39}$$

As shown in the proof of Theorem 1, equation 35–equation 37 imply

$$\bar{o}_{IF}^\pm = \pm\frac{\theta^\pm}{T} \left( \frac{T\bar{x}_{IF}^\pm}{\theta^\pm} + \frac{v^\pm(0)}{\theta^\pm} - \frac{v^\pm(T)}{\theta^\pm} \right) = \pm\frac{\theta^\pm}{T} Z^\pm, \tag{40}$$

where $\bar{x}_{IF}^\pm$ and $\bar{o}_{IF}^\pm$ denote the averages of $x_{IF}^\pm(t)$ and $o_{IF}^\pm(t)$ over $1 \leq t \leq T$, respectively, and $Z^\pm \in \mathbb{Z}$.

Combining equation 17, equation 39, and equation 40, we obtain

$$
\begin{aligned}
o_M &= \frac{\theta^\pm}{T} \left\lfloor \frac{T\bar{x}_{IF}^+}{\theta^+} + \frac{v^+(0)}{\theta^+} - \frac{T\bar{x}_{IF}^-}{\theta^+} \right\rfloor \\
&= \frac{\theta^+}{T} Z^+ + \frac{\theta^+}{T} \left\lfloor \frac{v^+(T)}{\theta^+} - \frac{T\bar{x}_{IF}^-}{\theta^+} \right\rfloor \\
&= \bar{o}_{IF}^+ + \frac{\theta^+}{T} \left\lfloor \underbrace{\frac{v^+(T)}{\theta^+} + \frac{v^-(0)}{\theta^+} - \frac{v^-(T)}{\theta^+}}_{\frac{C_v}{\theta^+}} - \frac{T\bar{o}_{IF}^-}{\theta^+} \right\rfloor,
\end{aligned}
\tag{41}
$$

where $C_v$ is a constant determined by the residual membrane potentials.

Therefore, the discrepancy satisfies

$$
\begin{aligned}
|\bar{o}_{IF} - o_M| &= \left| \bar{o}_{IF}^- + \frac{\theta^+}{T} \left\lfloor \frac{C_v}{\theta^+} - \frac{T\bar{o}_{IF}^-}{\theta^+} \right\rfloor \right| \\
&\leq \frac{|C_v| + \theta^+}{T}.
\end{aligned}
\tag{42}
$$

This establishes equation 34; moreover, by equation 5 each constant admits a reasonably small, $T$-independent upper bound.

## B   ABLATION STUDIES

**Impact of Scaling Factor**   Figure 3a illustrates the impact of the scaling factor $\lambda$ on both accuracy and energy ratio when applying ViT-Base SFormer to ImageNet-1K. As $\lambda$ increases from 0, accuracy rapidly improves and reaches its peak around $\lambda = 0.25$, after which it declines sharply. In contrast, the energy ratio decreases monotonically as $\lambda$ increases. This trade-off highlights the critical role of $\lambda$ in balancing task performance and computational cost. Particularly, our final choice of $\lambda = 0.252$ achieves a favorable compromise between accuracy and energy consumption.

To further support the effectiveness of this selection, additional visualizations in Appendix B depict how activations at the same location are converted under different values of $\lambda$. These results confirm that our method adaptively preserves information while enforcing spike sparsity.

We further analyze the impact of the normalization scale $p$ on model performance and energy efficiency. Detailed ablation results and discussions are provided in Appendix C.

**Impact of Normalization Scale**   Table 6 presents the ablation results of different normalization scale $p$ on ImageNet-1K. The best trade-off between accuracy and energy efficiency is observed at $p = 1.0$, which serves as our default setting. In particular, although $p = 0$ is theoretically expected to yield the highest accuracy, actual performance drastically decreases to 0.2%. This discrepancy stems from the highly uneven distribution of activations: a very small number of extreme values dominate,

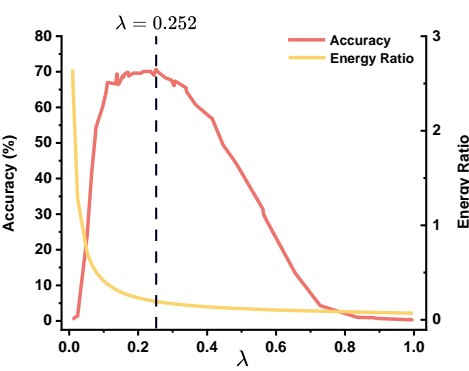 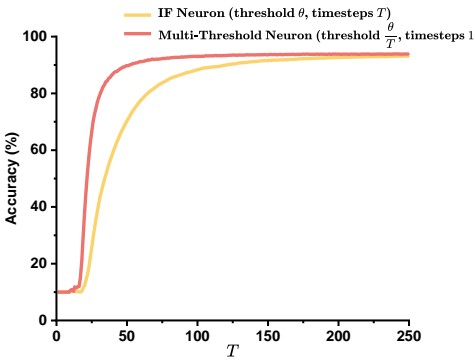

(a) Accuracy and energy ratio under different scaling factor $\lambda$ on ImageNet-1K (ViT-Base). The optimal $\lambda = 0.252$ is obtained via Bayesian optimization, balancing accuracy and energy ratio.

(b) Accuracy under different $T$ on CIFAR-10 using ResNet-18. The optimized Multi-Threshold Neuron with maximum $N = T$ achieves comparable or even superior performance to the IF Neuron.

Figure 3: (a) Scaling factor study and (b) equivalence validation.

Table 6: Ablation study on the impact of the normalization scale $p$ on ImageNet-1K using ViT-Base. Both accuracy and energy ratio are reported, with relative changes shown in parentheses relative to $p = 1.0$.

| $p$ | Accuracy (%) (relative to $p = 1.0$) | Energy Ratio (relative to $p=1.0$) |
|---|---|---|
| 0.0 | 0.2 (-99.7%) | 0.16 (-5.8%) |
| 0.5 | 63.7 (-9.9%) | 0.15 (-11.8%) |
| **1.0** | **70.7** | **0.17** |
| 1.5 | 70.8 (+0.1%) | 0.20 (+17.6%) |
| 2.0 | 71.2 (+0.7%) | 0.21 (+23.5%) |
| 2.5 | 70.9 (+0.2%) | 0.22 (29.4%) |

while the majority of activations are near zero. Consequently, spike rates become excessively sparse, resulting in insufficient firing activity to drive downstream layers effectively. This sparsity severely impairs information propagation, especially under the single-timestep setting ($T$=1), where it is difficult to compensate for missing spikes. On the other hand, increasing $p$ beyond 1.0 slightly improves accuracy but incurs a notable increase in energy consumption. These results validate the effectiveness of our chosen normalization scale.

**Impact of Number of Thresholds** We further study how the number of thresholds $N$ affects the accuracy–energy trade-off by varying $N \in \{8, 16, 32, 64\}$ and reporting accuracy and energy ratio for $1 \le T \le 4$ (7). Increasing $N$ consistently improves accuracy for all $T$, as a larger number of thresholds provides a finer approximation to the ANN activations. The improvement is especially pronounced when moving from $N = 8$ to $N = 16$ and then to $N = 32$. For example, at $T = 1$ the accuracy increases from $3.0\% \rightarrow 31.6\% \rightarrow 70.6\%$, while the corresponding energy ratios only grow from $0.09 \rightarrow 0.11 \rightarrow 0.17$.

However, the gain quickly saturates beyond $N = 32$. This reflects the fact that a larger $N$ introduces more comparisons and AC operations inside SFN, which linearly increases the energy but yields only limited additional representational benefit once the thresholds are sufficiently dense.

Considering both accuracy and energy, $N = 32$ offers the best trade-off. It achieves most of the accuracy across all $T$ while keeping the energy ratio substantially lower. Therefore, we adopt $N = 32$ as the default configuration in all main experiments.

Table 7: Impact of the number of thresholds $N$ on accuracy (%) and energy ratio for different time steps $T$ on the ViT-Base ImageNet-1K experiment. $N = 32$ provides the best accuracy–energy trade-off.

| $N$ | Accuracy/Energy | $T = 1$ | $T = 2$ | $T = 3$ | $T = 4$ |
|---|---|---|---|---|---|
| $N = 8$ | Acc. (%) | 3.0 | 12.7 | 21.7 | 27.3 |
| | Energy Ratio | 0.09 | 0.23 | 0.38 | 0.52 |
| $N = 16$ | Acc. (%) | 31.6 | 61.4 | 68.8 | 71.4 |
| | Energy Ratio | 0.11 | 0.27 | 0.44 | 0.60 |
| $N = 32$ | Acc. (%) | 70.6 | 77.6 | 79.3 | 80.0 |
| | Energy Ratio | 0.17 | 0.46 | 0.70 | 0.96 |
| $N = 64$ | Acc. (%) | 74.1 | 78.4 | 79.6 | 80.2 |
| | Energy Ratio | 0.24 | 0.53 | 0.82 | 1.11 |

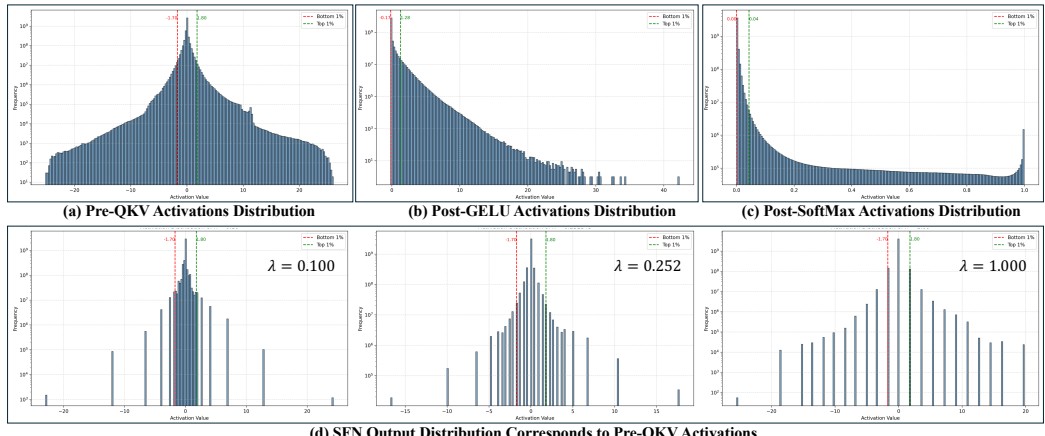

Figure 4: Activation and spike output distributions on ImageNet-1K using ViT-Base. (a)–(c): Distributions of activations from three stages in the ViT backbone — pre-QKV, post-GELU, and post-SoftMax, respectively; (d): Output spike distributions generated by the proposed SFN under different scaling factors $\lambda$, where the input is identical to (a). When $\lambda = 0.252$, the output distribution closely aligns with the original pre-QKV activation distribution, confirming the effectiveness of the selected scaling in preserving information structure during conversion.

## C  VISUALIZATION OF ACTIVATION AND SPIKE DISTRIBUTIONS

To further validate the effectiveness of our scaling strategy, we visualize the distributions of intermediate activations and the corresponding SFN spike outputs. Figure 4 (a)–(c) illustrate the activation distributions at three critical stages in the ViT-Base backbone: pre-QKV, post-GELU, and post-SoftMax. We observe that the activations exhibit a long-tailed, asymmetric distribution, which motivates our normalization strategy.

Figure 4 (d) shows the output spike distributions produced by the proposed SFN under different scaling factors $\lambda$, with the input fixed as the pre-QKV activations. When $\lambda$ is too small (e.g., 0.1), the distribution becomes overly concentrated near zero, leading to excessive sparsity. In contrast, a large $\lambda$ (e.g., 1.0) results in overly dispersed outputs that fail to match the original structure. Remarkably, $\lambda = 0.252$ yields a spike distribution that closely approximates the original activation distribution, demonstrating that our method adaptively preserves the information structure while promoting spike sparsity.

Table 8: Experimental configurations for various tasks.

| Task | Dataset | Hardware | Model Architecture | $\theta^{\text{softmax}}$ | $\lambda$ |
|------|---------|----------|-------------------|----------|----|
| Image Classification | ImageNet-1K | 1× NVIDIA V100 (32GB) | ViT-Base | 0.0035 | 0.252 |
| | | | ViT-Large | 0.0036 | 0.313 |
| | | | EVA | 0.0035 | 0.360 |
| Object Detection | COCO-2017 | 1× NVIDIA H20 (NVLink, 96GB) | Detectron2+EVA | 0.0036 | 0.350 |
| Instance Segmentation | COCO-2017 | 1× NVIDIA H20 (NVLink, 96GB) | Detectron2+EVA | 0.0036 | 0.333 |

## D  APPROXIMATE EQUIVALENCE VALIDATION

We evaluate the practical, error-bounded equivalence predicted by **Theorem 3**. On CIFAR-10 (Krizhevsky et al., 2009) with ResNet-18 (He et al., 2016), we compare a multi-timestep IF realization with a single-timestep multi-threshold realization. In both settings, the SNN is obtained by replacing ReLU with the corresponding spiking neuron.

As shown in Figure 3b, when $T < 25$, both neurons attain similarly low accuracy due to large conversion errors. As $T$ increases, the accuracy of both improves and their discrepancy gradually diminishes, consistent with the error-bounded approximate equivalence in Theorem 3. Across the range, the multi-threshold neuron consistently surpasses IF, indicating that the bounded discrepancy in practice shifts its output closer to the underlying ANN activation, thereby yielding higher accuracy. These results support the error-bounded equivalence and highlight the practical advantage of the proposed formulation for accurate, low-latency SNN inference.

## E  IMPLEMENTATION DETAILS

### E.1  TASK-SPECIFIC SETTINGS

Table 8 summarizes the task-specific settings for image classification, object detection, and instance segmentation. For each dataset and model architecture, we report the post-SoftMax threshold parameter $\theta^{\text{softmax}}$ and the scaling factor $\lambda$ used in our SFN.

### E.2  BAYESIAN OPTIMIZATION FOR SCALING FACTOR

The scaling factor $\lambda$ is optimized via Bayesian optimization (`n_trials`=50) on a validation split formed by a random 2% subsample of the dataset. Consequently, the total tuning overhead is approximately one full evaluation ($50 \times 2\% = 100\%$).

### E.3  EVALUATION METRICS AND ENERGY METRICS

We adopt standard evaluation metrics for each task. For image classification, we report top-1 accuracy. For object detection and instance segmentation, we follow the COCO evaluation protocol and report mAP@0.5 and mAP@[0.5:0.95].

For energy efficiency evaluation, we use the Energy Ratio, defined as the ratio of total energy consumption between the SNN and its corresponding ANN. Following Rathi & Roy (2020); Huang et al. (2024), the Energy Ratio is computed as:

$$\text{Energy Ratio} = \frac{\#\text{AC}_{\text{SNN}} \times E_{\text{AC}} + \#\text{MAC}_{\text{SNN}} \times E_{\text{MAC}}}{\#\text{MAC}_{\text{ANN}} \times E_{\text{MAC}}}, \tag{43}$$

where $\#\text{AC}_{\text{SNN}}$ and $\#\text{MAC}_{\text{SNN}}$ denotes the number of additions and multiplications in SNN, respectively. And $\#\text{MAC}_{\text{ANN}}$ is the number of additions in ANN. We follow prior work (Horowitz, 2014) and set the energy cost per operation as $E_{\text{AC}} = 0.9pJ$ and $E_{\text{MAC}} = 4.6pJ$.

In SNNs, spikes are discrete, thus the dominant operations are additions. Specifically, integer multiplications are realized by repeated additions, i.e. $w \cdot n = \underbrace{w + \cdots + w}_{n \text{ times}}$. Consequently, we

approximate equation 43 as:

$$\text{Energy Ratio} \approx \frac{\#\text{AC}_{\text{SNN}} \times 0.9pJ}{\#\text{MAC}_{\text{ANN}} \times 4.6pJ}. \tag{44}$$

## F  THE USE OF LARGE LANGUAGE MODELS

We used an LLM solely to aid and polish writing (wording, grammar, and style), including minor edits for clarity and concision, rephrasing author-written passages, harmonizing terminology/notation, and light LaTeX formatting (e.g., captions and tables). The LLM did not contribute to research ideation, model design, experiments, data analysis, or result interpretation. All technical content, equations, and claims were authored and verified by the authors.

