# OpenReview forum: "One-Timestep is Enough: Achieving High-performance ANN-to-SNN Conversion via Scale-and-Fire Neurons"
_ICLR.cc/2026/Conference — Submitted to ICLR 2026_

### Official Review · Reviewer_Q445 · 2025-10-18

**Soundness:** 3
**Presentation:** 2
**Contribution:** 2
**Rating:** 4
**Confidence:** 4

**Summary:**

This paper proposes a new ANN-to-SNN conversion framework called **Scale-and-Fire Neuron (SFN)**, which enables **single-timestep (T=1)** inference while maintaining high accuracy. The authors introduce the **Temporal-to-Spatial Equivalence Theory**, which proves that multi-timestep integrate-and-fire (IF) neurons can be equivalently replaced by single-timestep **multi-threshold neurons (MTN)**.

Building upon this theory, the SFN integrates:
1. A **membrane potential scaling mechanism**, where the **scaling factor (λ)** is optimized via **Bayesian optimization** to balance spike sparsity and accuracy.
2. An **adaptive fire function**, whose discrete thresholds $ \theta_i $ are *determined* based on the activation distribution and are proportional to the optimized scaling factor ($ \theta_i = \lambda \theta y_i $).

The authors extend this design to Transformer architectures, forming the **SFN-based Spiking Transformer (SFormer)**, which aligns spike distributions with attention patterns. Experiments show strong performance across multiple vision benchmarks:
- **ImageNet-1K:** 88.8% Top-1 accuracy at $T = 1$.
- **COCO-2017 Detection:** 78.2% mAP@0.5.
- **Energy efficiency:** up to 81% reduction in energy consumption relative to the ANN baseline.

Overall, the paper provides both a theoretical and practical framework for **high-performance single-timestep ANN-to-SNN conversion**, bridging temporal spiking integration and spatial multi-threshold encoding.

**Strengths:**

The paper introduces the **Scale-and-Fire Neuron (SFN)** and the **Temporal-to-Spatial Equivalence Theory**, enabling single-timestep ($T=1$) ANN-to-SNN conversion with high accuracy. Experiments on ImageNet and COCO demonstrate good empirical performance and energy savings. The integration of a Bayesian-optimized scaling factor ($\lambda$) and adaptive multi-threshold firing is both elegant and effective.

The work is clearly written. It shows that high-performance ANN-to-SNN conversion can be achieved without multi-timestep accumulation. The framework’s applicability to Transformer architectures (SFormer) further enhances its relevance to large-scale vision models and energy-efficient AI research.

**Weaknesses:**

1) Originality: the Temporal-to-Spatial Equivalence is new, but the practical recipe (multi-threshold neurons at single step $T=1$ with scaling $\lambda$ and density-aware thresholds) overlaps with prior multi-threshold or dynamic-threshold conversion (Huang et al., 2024; Li et al., 2025; MT-SNN). The paper does not show a clear win at matched $N$, nor operator-aware bounds beyond the theory’s non-negative/linear assumptions.

2)  SoftMax is explicitly handled via a max-driven cap $\theta_{\text{softmax}}$; GELU is only implicitly covered (no operator-level rule/ablation); LayerNorm handling is missing (no pre/post-LN placement, no treatment of signed activations). This is a material reproducibility and validity gap for ViT-style models.

3) Practical efficiency evidence is incomplete
The method trades time for spatial threshold multiplicity. Per-neuron updates scale with $N$, and spike/event traffic can grow with $N$. No $N$-sweep or absolute energy (mJ/inference), and no accuracy-matched multi-timestep IF ($T>1$) baseline with end-to-end latency/energy and per-layer spike histograms.

4) Cost/justification of $\lambda$
$\lambda$ is selected by Bayesian optimization; the search budget/range/compute cost are undisclosed, and there is no comparison to simple analytical or percentile estimators, weakening the “training-free” claim.

5) Sensitivity and robustness
A max-driven $\theta_{\text{softmax}}$ is outlier-sensitive; there is no percentile-based alternative or drift analysis. Stability of density-aware thresholds under activation shifts is also unreported.

6) Large $N$ can be counterproductive
Energy and event counts tend to rise with $N$; neuron-side work grows $O(N)$ and can erode the latency gain; hardware burden (threshold storage/lookup, event queues) increases; fine binning can overfit activation noise and hurt generalization; returns diminish once $N$ already covers the bulk of the activation mass (often around a moderate value like 32). The chosen $N$ should be justified via an $N$-sweep (latency/energy/event counts/bandwidth/accuracy) and a fair $T>1$ baseline.

References

Huang, X. et al., 2024. “Towards High-Performance Spiking Transformers from ANN to SNN Conversion.” https://arxiv.org/pdf/2502.21193
Li, Y. et al., 2025. “Multi-Threshold Neuron Models for Single-Step ANN-to-SNN Conversion.” https://arxiv.org/pdf/2503.00301
Wang, Z. and Zhang, T., 2023/2024. “MT-SNN: Enhance Spiking Neural Network with Multiple Thresholds.” https://arxiv.org/pdf/2303.11127
Fan, Y. et al., 2025. “A multisynaptic spiking neuron for simultaneously encoding spatiotemporal dynamics.” Nature Communications. https://www.nature.com/articles/s41467-025-62251-6

**Questions:**

Q1. Originality and theoretical contribution
The Temporal-to-Spatial Equivalence theory is interesting, but its practical form (multi-threshold + λ-scaling + adaptive thresholds) resembles prior MT-SNN or dynamic-threshold methods (Huang 2024; Li 2025). Could the authors clarify what is fundamentally novel in Scale-and-Fire Neuron?

Q2. Theorems 1–2 assume non-negative inputs and linear operations, which may not hold with GELU, LayerNorm, or signed activations.
Can the authors quantify the impact of these violations and explain how such nonlinearities are handled or approximated in practice?

Q3. Scalability and applicability
Table 1 shows O(N) neuron cost, but experiments fix N = 32.
Can the authors include an N-sweep (e.g., 8–64) showing accuracy, latency, and energy trade-offs?

---

> ### Author Response · Authors · 2025-11-24
> **Reply to Reviewer Q445**
>
> 1. **Novelty of SFN compared with other MTN methods**:
>
> The main difference between our SFN and existing MTN methods lies in the explicit emphasis on the scaling factor. In other MTN methods, there is no notion of such a scaling factor. Theoretically, the Temporal-to-Spatial Equivalence theory characterizes that the scaling factor should be proportional to the minimal non-degenerate time steps T required by the model; in our ablation studies, Table 5 further shows that the scaling factor has a significant impact on performance at T=1. To the best of our knowledge, existing MTN designs invariably become highly degraded in the single-step regime.
>
> In addition, SFN adopts a flexible sparse firing mechanism, which is also considered in [1]. The difference is that [1] uses a fixed exponentially spaced firing scheme, while SFN designs a more suitable firing pattern by explicitly aligning the activation distribution. We compare several firing schemes in Table 5 (“linear” corresponds to most MTN methods, “exponential” corresponds to [1]).
>
> 2. **Handling of nonlinear components**:
>
> For T>1, we adopt ECM [1] to handle the nonlinear components in the model (GeLU, LayerNorm, Softmax, and the QK matrix multiplication). ECM ensures input–output linearity along the temporal dimension, so that when the network runs asynchronously (step by step), the activations at each layer are automatically aligned over time. Our released code includes the corresponding treatment of these nonlinear components, making our results fully reproducible.
>
> 3. **Handling negative activations**:
>
> Theorem 3 introduces dual neurons with positive and negative thresholds to handle negative activations and establishes an “approximate equivalence” that is slightly weaker than strict equivalence.
>
> 4. **Energy analysis and additional experiments over N**:
>
> Increasing N introduces more comparison operations inside SFN. These are pointwise operations, and the energy cost of a single comparison is typically no higher than that of a single AC, so their contribution to the overall model energy is very small compared with the whole network.
>
> Our models are simulated on GPUs rather than real neuromorphic hardware, we follow [1] and report the energy ratio. We approximate energy by the AC/MAC counts, as nonlinearities, comparisons, and pointwise multiplications are negligible compared with linear layers. In this ratio, we explicitly account for repeated AC operations caused by multiple spike emissions. Table 4 shows that the SNNs converted by our method remain energy-efficient.
>
> Furthermore, we have added comparison experiments for different N (Table 7) in our revised paper, reporting accuracy and energy ratios for 1≤T≤4. The results show that, as N increases, both accuracy and energy ratio increase under the same T. Considering the accuracy–energy trade-off at T=1, we ultimately choose N=32.
>
> 5. **Search cost for the scaling factor**:
>
> Appendix E.2 describes the search cost for the scaling factor. In addition, Table 7 shows that this factor lies in a similar range across different model–task combinations, and Fig. 3(a) indicates that performance remains high within a reasonable range of values. Therefore, if one wishes to avoid this additional search cost, directly setting λ=0.3 is also a practical strategy.
>
> 6. **Sensitivity of the softmax thresholds**:
>
> Using the maximum value may indeed be sensitive to a small number of outliers. To improve robustness, we will replace the max-based choice with a top-k based selection for the softmax thresholds.
>
> **References**:
>
> [1] Huang, et al. “Towards High-Performance Spiking Transformers from ANN to SNN Conversion.”

---

> ### Author Response · Authors · 2025-11-28
>
> We would like to sincerely thank you again for your constructive comments. In the revised manuscript and rebuttal, we have carefully addressed each of the raised points. We hope that these clarifications and additional results help to resolve your main concerns, and we would very much appreciate it if you could let us know whether the current revision adequately addresses your previous comments.

---

### Official Review · Reviewer_aVto · 2025-10-28

**Soundness:** 1
**Presentation:** 2
**Contribution:** 1
**Rating:** 2
**Confidence:** 5

**Summary:**

This paper proposes SFN, which enables converted SNNs to achieve good performance within 1 time step.

**Strengths:**

The proposed SFN enables converted SNNs to achieve good performance with only 1 timestep.

**Weaknesses:**

1. The proposed Scale-and-Fire Neuron (SFN) largely mirrors the non-uniform activation quantization with calibration used in quanted ANNs.  The so-called "Temporal-to-Spatial Equivalence Theory" is quite obvious and superficial. The resulting SFN actually transmits floating-point values and relies heavily on intricate searches for an appropriate scaling factor, which is unlikely to be a fixed constant that is completely task-independent and model-independent (as also acknowledged by the authors in Section 3.2.2), undermining the purported motivation and simplicity of spiking/neural dynamics.

2. This work proposed to replace "multi-step membrane potential integration" with "single-step multi-threshold", aiming to transform temporal integration into spatial multi-level thresholds. However, this eliminates neural dynamics (membrane potential evolution and temporal correlation of residual membrane potential) with spatial quantized intervals. The resulting SFN is more of an engineering parameterization of step interval and threshold assignment, and is unrelated to neural dynamics.

3. To achieve better performance, the authors stacked multiple heuristics: positive and negative branches, specialized upper threshold bounds for the SoftMax layer, Bayesian optimization of λ, quantile p, and varying threshold densities at each level. These are all a combination of a posteriori calibration properties, resulting in increased implementation complexity and parameter sensitivity, undermining the simple temporal processing mechanism that SNNs are supposed to possess. In contrast, a clean IF/BPTT or reliable ANN quantization scheme is more maintainable and verifiable.

4. The purpose of ANN-to-SNN conversion is to convert the pretrained ANN into an SNN to take advantage of the high performance of ANNs and the high energy efficiency of SNNs.  However, the proposed SFN cannot actually be converted into a single/simple IF/LIF model during inference, and even complicates the membrane potential accumulation process of the spiking neuron. In fact, during this converted SNN inference, the computational complexity of each operation is not just O(1). Therefore, it is unreasonable to estimate the energy consumption of 0.9pJ per synaptic operation as in previous literature.

5. SFN is not a standard neuron. Multiple threshold comparisons, non-uniform threshold mapping, positive and negative branching, and dedicated SoftMax upper bounds all introduce control flow and table lookup overhead; implementing this type of piecewise non-uniform thresholding is not "free." The energy consumption metrics used in this paper are still approximated by an operator energy model (treating multiplication as multiple additions), without actual end-to-end chip or FPGA-level simulations/measurements or timing analysis. More importantly, while single-step execution with N thresholds reduces the repetition of non-neuronal operators, it significantly increases the number of firing spikes and additional operations, e.g., comparisons per layer when N is large, potentially offsetting the energy/latency gains. Overall, SFN is not a standard neuron, while the paper does not provide the corresponding hardware deployment strategy and energy consumption/delay analysis. Therefore, the provided analysis results at the operation-level using normal IF/LIF models are obviously not rigorous.

**Questions:**

1. SFN requires a lot of hyperparameters. Can they have a certain degree of generalization across different tasks/models?

2. SFN actually transmits a floating-point value. What's the actual difference between SFN and quantized ANN activation? Running a quantized ANN only requires one timestep, while it does not need that many complicated hyperparameters or additional runtime computations(e.g., comparisons of positive/negative values).
If the author cannot provide the hardware implementation strategy or analysis, what are the actual benefits of SFN compared to mature quantized ANNs that can run on dozens of well-commercialized hardware?

3. It seems that SFN aims to force the firing rate of the SNN to align with the activation value of each layer of the ANN. If T is not 1, can it run step-by-step(asynchronously) rather than layer-by-layer(synchronously)?

4. For SNNs, a single time step is almost a degenerate case. This means no stateful units, no time integration, and no dynamics. I am curious whether the proposed SFN can work well on neuromorphic datasets or why does the author consider SFN to be classified as a spiking neuron? Clearly, the values ​​transmitted by SFN are not binary spikes, cannot convert MAC to AC, are not energy-efficient, and layer-by-layer calibration would prevent the network from operating asynchronously.

---

> ### Author Response · Authors · 2025-11-24
> **Reply to Reviewer aVto**
>
> 1. **Hyperparameter issues and differences between single-step SNNs and quantized ANNs**:
>
> We provide detailed explanations regarding the hyperparameter issues and the differences between single-step SNNs and quantized ANNs in the common comment. In summary, the hyperparameters have been carefully tuned to ensure stable performance and generalization across different models and tasks, and our single-step SNNs offer significant energy efficiency over quantized ANNs, with the ability to extend to T>1.
>
> 2. **Transmitted values in SFN**:
>
> In GPU simulation, SFN transmits a floating-point value and performs MAC. Conceptually, however, SFN emits integer spikes, and the corresponding floating-point output is simply the product of the spike count and the SFN floating-point threshold. This product can be viewed as a fixed-point scaling, which can either be absorbed into the weights of the subsequent linear layer or implemented efficiently. Integer-valued spike neurons of this form are also common in existing SNNs, such as ECMT [1], I-LIF [2], MT-SNN [3] and [4].
>
> 3. **Hardware implementation**:
>
> In practice, our results are obtained via GPU-based simulation and an operation-count–based energy model. Nevertheless, as multi-threshold designs [1], [2], [3], [4] become more widely adopted in the SNN community, we expect that energy-efficient multi-threshold neuromorphic hardware capable of realizing such AC-based implementations will emerge in the future.
>
> Deployment of the proposed SFN on such neuromorphic hardware enables the realization of high energy efficiency. This is because the energy cost of pointwise comparisons is negligible compared to AC/MAC operations. Additionally, since most neurons fire relatively small spike counts (Figure. 4), the use of AC instead of MAC operations will further reduce energy consumption.
>
> 4. **Asynchronously execution for T > 1**:
>
> SFN is designed to align activations at each layer, but this alignment does not have to be enforced layer by layer. For T > 1, we adopt ECM [1] to handle nonlinear components, so that remains linear along the temporal dimension and satisfies Theorem 2. This directly enables step-by-step asynchronous execution. More concretely, the temporal dynamics of a single SFN ensure that its average firing rate over a time window matches the average input activation. The linearity along temporal dimension guaranteed by ECM [1] then propagates this alignment from one layer to the next. As a result, even when the network executes along the temporal dimension (step by step), it still preserves activation alignment layer by layer.
>
> 5. **SFN as a spiking neuron**:
> - SFN can be extended along the temporal dimension and improves performance.
> - SFN emits integer spikes, like other MTN such as [1], [2], [3], [4].
> - In ANN2SNN works (e.g., [5] [6] [7]), neuromorphic datasets are rarely used, even for multi-step conversion, because they are typically used to train SNNs directly, and suitable ANN baselines with pretrained weights are scarce.
>
> 6. **Computational complexity and overall model-level energy evaluation**:
>
> We will include it in the supplement to this reply.
>
> 7. **Summary of the necessity and goals of our work**:
>
> We will include it in the supplement to this reply.
>
> **References**:
>
> [1] Huang, et al. “Towards High-Performance Spiking Transformers from ANN to SNN Conversion.”
>
> [2] Luo, et al. “Integer-Valued Training and Spike-Driven Inference Spiking Neural Network for High-Performance and Energy-Efficient Object Detection.”
>
> [3] Wang et al., “MT-SNN: Enhance Spiking Neural Network with Multiple Thresholds.”
>
> [4] Xu et al., “Direct Training via Backpropagation for Ultra-Low-Latency Spiking Neural Networks with Multi-Threshold LIF.”
>
> [5] Rueckauer et al., “Conversion of Continuous-Valued Deep Networks to Efficient Event-Driven Networks for Image Classification.”
>
> [6] Deng & Gu, “Optimal Conversion of Conventional Artificial Neural Networks to Spiking Neural Networks.”
>
> [7] Ding et al., “Optimal ANN-SNN Conversion for Fast and Accurate Inference in Deep Spiking Neural Networks.”

---

> > ### Comment · Reviewer_aVto · 2025-11-24
> >
> > Thanks to the authors for the rebuttal, but unfortunately, many of my concerns remain unaddressed.  The author did not provide results on neuromorphic benchmarks. The proposed SFN needs **dynamic threshold**, and while the author claimed "this product can be viewed as a fixed-point scaling", it in fact cannot be mapped to standard IF/LIF, resulting in floating-point operations. The **assumption** regarding energy savings relies on the premise of feasible hardware deployment, and mainly because the authors count ANN operation as 4.6pJ/OP and SNN operation as 0.9pJ/OP; this "0.9pJ/OP" comes from the evaluation on Loihi1 with 0/1 spike. Clearly, this does not apply to SFNs.
> >
> > Please provide hardware implementation and simulation results instead of continuing to claim the benefits of your method. There are many open-source tools available, and thus no excuses.
> >
> > I keep my rating.

---

> > > ### Author Response · Authors · 2025-11-24
> > >
> > > Thank you for your valuable feedback. We understand your concerns regarding the lack of hardware implementation and simulation results with SFN. To address the point you raised about hardware implementation, could you clarify which open-source tools you are referring to for evaluating neuromorphic models? This will help us align our work with the appropriate tools and approaches.
> > >
> > > Additionally, while we acknowledge the importance of hardware evaluations, we would like to point out that many works employing MTN or dynamic threshold have not been tested on neuromorphic hardware either.
> > > - MT-SNN [1], LM-HT [2], I-LIF [3]: MTN methods;
> > > - ECMT [4]: MTN methods also requiring dynamic threhsolds;
> > > - MSAT [5]: Dynamic threhsold model.
> > >
> > > Implementing MTN or SFN on neuromorphic hardware is a complex and resource-intensive task.
> > >
> > > **References**:
> > >
> > > [1] Wang et al., “MT-SNN: Enhance Spiking Neural Network with Multiple Thresholds.”
> > >
> > > [2] Hao et al., “LM-HT SNN: Enhancing the Performance of SNN to ANN Counterpart through Learnable Multi-hierarchical Threshold Model.”
> > >
> > > [3] Luo, et al. “Integer-Valued Training and Spike-Driven Inference Spiking Neural Network for High-Performance and Energy-Efficient Object Detection.”
> > >
> > > [4] Huang, et al. “Towards High-Performance Spiking Transformers from ANN to SNN Conversion.”
> > >
> > > [5] He et al., “MSAT: Biologically Inspired Multi-Stage Adaptive Threshold for Conversion of Spiking Neural Networks.”

---

> ### Author Response · Authors · 2025-11-24
> **Supplement to the Reply to Reviewer aVto**
>
> 6. **Computational complexity and overall model-level energy evaluation**:
>
> For a single SFN, the computational complexity is O(N), as summarized in Table 1. However, at the level of the whole network, the threshold comparisons have limited impact on energy: they are pointwise operations, and the cost of a single comparison is typically no higher than that of one AC, so their contribution can be neglected. SFN does increase the number of emitted spikes, and we explicitly account for this by counting the repeated AC operations induced by multi-spike firing. The estimated AC energy of the model is given by the total number of AC operations multiplied by the per-AC cost (0.9 pJ).
>
> Table 1 also highlights another advantage of single-step SNNs: they avoid repeated evaluations of nonlinear modules, which is consistent with standard ANN inference. Consequently, the dominant difference in energy between the converted SNN and the original ANN comes from converting MAC to AC.
>
> As noted in (3), our models are simulated on GPUs, and we follow common practice [1] [2] [3] and report the energy ratio. We approximate energy by the AC/MAC counts, since nonlinear components are shared between ANN and SNN, and the overhead of SFN operations and other pointwise computations is negligible compared with the matrix multiplications in linear layers. In this AC/MAC-based ratio, we explicitly account for repeated AC operations caused by multiple spike emissions (Appendix E.3). Table 4 shows that, under this metric, the SNNs converted by our method remain energy-efficient.
>
> 7. **Summary of the necessity and goals of our work**:
>
> Our goal is to propose an ANN2SNN method that not only performs well for T>1, like existing approaches, but also maintains strong performance at T=1, thereby reducing inference latency. Our proposed Temporal-to-Spatial Equivalence theory provides a unified view of multi-threshold neurons and scaling factors: the thresholds and scaling interact with timesteps. This explains why previous ANN2SNN methods inevitably degenerate in the T=1 regime, and provides an SNN framework that, at T=1, behaves like a PTQ-style quantizer while still enabling MAC-to-AC conversion, and, when extended to T>1, preserves genuine spiking temporal dynamics.
>
> **References**:
>
> [1] Huang, et al. “Towards High-Performance Spiking Transformers from ANN to SNN Conversion.”
>
> [2] Wang et al., “Signed Neuron with Memory: Towards Simple, Accurate and High-Efficient ANN-SNN Conversion”
>
> [3] Li et al., “Efficient and Accurate Conversion of Spiking Neural Networks”

---

> ### Comment · Reviewer_aVto · 2025-11-25
>
> My concerns actually extend far beyond hardware implementation/simulation.
>
> For the point "energy saving", (4.6-0.9)/4.6 ≈0.8, directly counts MACs as ACs can already lead to 80% energy saving; that's what I see in this paper, which means the transmitted value is not sparse, but dense. Besides, I must say the 4.6pJ/OP energy is also evaluated on Loihi1, fabricated using Intel's 14 nm process. Lower power consumption has been achieved through dedicated quantized ANN accelerators.
>
> For the point "asynchronous execution for T > 1" and "hardware implementation", I do not buy this plain claim without experimental verification.
>
> For the point "lack of pretrained weights on neuromorphic benchmarks", research on event vision in ANNs began earlier than that in SNNs, and the datasets were generally smaller. The lack of a suitable pre-trained network is not a sufficient reason. Besides, for your implementation, please do not try to replicate the "whole"  input event stream multiple times just to achieve a conversion; that's meaningless.
>
> For the works cited by the authors:
>
> [1] arxiv 2023, not yet accepted/published;
>
> [3] ECCV 2024, a fake SNN but true quantized ANN that directly transmits quantized activation, though the authors of [3] named that as "firing rate"/"integer-valued training". To this day, how I-LIF neurons are converted into true SNN neurons during inference remains a mystery. Forgive my frankness, but the SFN in this article is doing the same thing as trying to first discretize the activation values ​​in ANNs, then directly transmit "firing rates"/xx value (in this work, "spatial multi-threshold") in so-called converted SNNs. Figure 4(d) is the direct evidence.
>
> [4] ACM-MM and [5] Neural Computing and Applications. I would consider rating this work 4 out of 10 for ACM-MM, or a major revision for a journal like Neural Computing and Applications, but a clear rejection for ICLR.
>
> [2] Neurips 2024, allowed transmission values ​​to be integer multiples of a fixed threshold. This can be implemented on Loihi2 with graded spikes, but at the expense of increased energy cost.
>
> The authors insist that their method can achieve energy efficiency gains on neuromorphic hardware, but seem unfamiliar with neuromorphic hardware and ask reviewers for available tools. The hint is that you can use separate tools to obtain effective mappings on commercially available neuromorphic hardware and assess power consumption. Please provide port connections for hardware simulation and the energy breakdown (static/dynamic).
>
> I will not respond to rebuttals until a new, strong response emerges that can change the facts, or until my peers provide new comments. Good luck.

---

### Official Review · Reviewer_37fc · 2025-11-01

**Soundness:** 2
**Presentation:** 2
**Contribution:** 2
**Rating:** 2
**Confidence:** 4

**Summary:**

This paper presents a novel framework for high-performance, single-timestep (T=1) Artificial Neural Network (ANN) to Spiking Neural Network (SNN) conversion. The authors introduce a "Temporal-to-Spatial Equivalence Theory" to formally connect multi-timestep Integrate-and-Fire (IF) neurons with single-timestep Multi-Threshold Neurons (MTN). Based on this theory, they propose the Scale-and-Fire Neuron (SFN) and an SFN-based Spiking Transformer (SFormer). In the end, the authors show a new SOTA on ImageNet-1K with 88.8% top-1 accuracy at T=1.

**Strengths:**

1. The authors attempt to provide a rigorous theoretical underpinning for their single-time step approach through the "Temporal-to-Spatial Equivalence Theory". Grounding the methodology in a formal equivalence (even under ideal conditions) is a commendable effort that adds depth and clarity to the proposed conversion framework.

2. The proposed Scale-and-Fire Neuron (SFN) is a well-motivated design. It moves beyond a naive multi-threshold implementation by incorporating a scaling factor (λ) and an adaptive firing function. This design directly addresses the practical challenges of converting large models where activation distributions can be highly skewed and varied. The use of Bayesian optimization to tune λ is a principled approach to finding a good balance between accuracy and spike sparsity.

**Weaknesses:**

1. In Table 2, the performance comparison between different architectures is obviously unfair, the author should compare with other ANN2SNN methods using the same model architecture
2. Since the model uses multi thresholds and the time step is 1 only, the model is more similar to an activation quantized only model. Therefore, the comparison with some typical quantization methods like [1], is also necessary.
3. The citations of all existing other methods on all performance comparison tables are missing.

[1] Elias Frantar, et al. GPTQ: Accurate Post-Training Quantization for Generative Pre-trained Transformers.

**Questions:**

None.

---

> ### Author Response · Authors · 2025-11-24
> **Reply to Reviewer 37fc**
>
> 1. **Performance comparison across architectures**:
>
> Among the ANN2SNN methods we surveyed, only ECMT [1] reports ImageNet-1K results using the same architecture, and we provide a detailed comparison with this method in Tables 2 and 4.
>
> We also add experiments applying SFN to VGG-16 on ImageNet-1K. Since the reported VGG-16 baselines differ slightly across works, we compare methods in terms of accuracy loss relative to their original ANNs for fairness. In our experiments, the converted SNN at T=1 reduces top-1 accuracy from 72.9% to 72.7% (a drop of 0.2%), which is smaller than the drop reported by the best existing method OPI (74.8% → 74.2%, a drop of 0.6%).
>
> 2. **Comparison with quantized ANN models**:
>
> We additionally report a comparison between our method and training-free activation-quantized ANNs on ImageNet-1K using ViT-Base. For fairness, we align the number of thresholds N with the activation quantization bit-width and report the accuracy drop relative to the original ANN. Specifically, N=8 corresponds to W3A3, and N=16 corresponds to W4A4.
>
> Table 1: Comparison of SNN with Original ANN in Terms of Accuracy Drop for Different N
> |  N   | Performance  | T=1 | T=2 | T=3 | T=4 |
> |  ----  | ----  | ----  | ----  | ----  | ----  |
> | N=8  | Acc drop (%) | -77.8 | -68.1 | -59.1 | -53.5 |
> |  | Energy Ratio | 0.09 | 0.23 | 0.38 | 0.52 |
> | N=16  | Acc drop (%) | -49.2 | -19.4 | -12.0 | -9.4 |
> |  | Energy Ratio | 0.11 | 0.27 | 0.44 | 0.60 |
>
> Table 2. Comparison of Quantized ANN with Original ANN in Terms of Accuracy Drop for Different Quantization Bits
> |  Method   | Bit (W/A)  | Acc drop (%) |
> |  ----  | ----  | ----  |
> | PTQ4ViT [2]  | 3/3 | -84.53 |
> |  | 4/4 | -53.85 |
> | RepQ-ViT [3]  | 3/3 | -84.4 |
> |  | 4/4 | -16.06 |
> | BRECQ [4]]  | 3/3 | -83.95 |
> |  | 4/4 | -74.86 |
> | APQ-ViT [5]  | 3/3 | ~ |
> |  | 4/4 | -43.13 |
> | AdaLog [6]  | 3/3 | -46.63 |
> |  | 4/4 | -4.86 |
>
> In the SNN, the weights remain 32-bit, while in the activation-quantized ANN, the weight bit-width is aligned with the activation bit-width (since there is no method for activation-only quantization while keeping the weights at 32-bit). Weight quantization has minimal impact on model performance, as shown in several studies [7] [8], which justifies that the mismatch between the weight and activation bit-widths in our experiments does not significantly affect performance.
>
> The results show that our single-step SNN outperforms all quantization methods except AdaLog [6] under W3A3, and surpasses a subset of quantization methods under W4A4. Moreover, by increasing the inference time steps, our method can approach the performance of the best quantization methods.
>
> Although a single-step SNN still falls short of the best quantized models in terms of accuracy, it is important to emphasize the energy-efficiency advantage of spike-based computation compared with activation quantization. We explain the detailed differences between our single-step SNNs and quantized ANNs in our common comment.
>
> 3. **Missing table references**:
>
> We have added the missing table references in the revised version of the paper.
>
> **References**:
>
> [1] Huang, et al. “Towards High-Performance Spiking Transformers from ANN to SNN Conversion.”
>
> [2] Yuan, et al. “PTQ4ViT: Post-training quantization for vision transformers with twin uniform quantization.”
>
> [3] Li, et al. “RepQ-ViT: Scale reparameterization for post-training quantization of vision transformers.”
>
> [4] Li, et al. “BRECQ: Pushing the limit of post-training quantization by block reconstruction.”
>
> [5] Ding, et al. “Towards accurate post-training quantization for vision transformer.”
>
> [6] Wu, et al. “Adalog: Post-training quantization for vision transformers with adaptive logarithm quantizer.”
>
> [7] Nagel, et al. "A White Paper on Neural Network Quantization."
>
> [8] Yao, et al. "ZeroQuant-V2: Exploring Post-training Quantization in LLMs from Comprehensive Study to Low Rank Compensation."

---

> > ### Comment · Reviewer_37fc · 2025-11-26
> > **Response to rebuttal**
> >
> > Thank you for your rebuttal. I have two remaining points that force me to keep my score as is.
> >
> > 1) When you go above T=1 you must account for the cost of memory accesses (reading activations / weights multiple times) which is not accounted for in the current energy models which seems to solely based on AC/MACs.
> >
> > 2) The cost of the SFN multiple thresholds is again focused on ACs with no cost of the additional control and memory access that other reviewers point to. While other papers also may indeed rely on energy estimates I believe for ICLR requires more comprehensive proof of energy efficiency claims.

---

### Official Review · Reviewer_pZCi · 2025-11-03

**Soundness:** 2
**Presentation:** 3
**Contribution:** 2
**Rating:** 4
**Confidence:** 5

**Summary:**

This paper aims to reduce the inference time step of ANN-to-SNN conversion to 1 time step, while maintaining high conversion accuracy. Specifically, it proves that under certain conditions, multi-time-step Integrate-and-Fire (IF) neurons are equivalent to one-time-step Multi-Threshold Neurons (MTN). Then, it proposes the Scaleand-Fire Neuron (SFN) model for use in the Transformer architecture to convert ANNs into SNNs with one-time-step.

**Strengths:**

The writing of this paper is fluent, with a well-structured organization. All the theories claimed in the paper have been properly illustrated and elaborated.

**Weaknesses:**

1. **Question on the correctness of Theorem 1**: Theorem 1 requires that "the input is bounded by θ", which constitutes a rather strong constraint. How to ensure this constraint can be satisfied? This is because the input is correlated with both input activations and weights.


2. **Clarification on the definition of variables and equivalence of outputs**: What is the meaning of \( o_M \) in Equation (27)? Is it consistent with the definition of \( o(t) \) in Equation (6)? It is noted that \( o_M \) has no temporal dimension, while \( o(t) \), although containing a temporal dimension, only represents the output at a single time step. In summary, please provide a detailed explanation of the following:
   - What exactly are the equivalent outputs of the MTN and the IF  model?  Is their equivalent output given by Equation (32)?
   - Additionally, since the entire paper focuses on **one time step**, the temporal dimension becomes meaningless. In this case, the work degrades to a simple activation quantization task. Consequently, the MTN—i.e., the model described in Equation (6), merely serves as an activation quantizer for artificial neural networks (ANNs).

3. **Question on the necessity of this work**: Theorem 2 is self-evident and does not require a separate proof, which raises doubts about the necessity of this study. For ANN-to-SNN conversion works, the core goal is to transform ANNs into SNNs. However, the essence of this work is an ANN-to-ANN conversion (the MTN  functions only as an activation quantization function and does not introduce temporal dimension).

4. **Further question on the necessity of this work**: Considering that this paper discusses one timestep, are the t in h(t) and o(t) in formulas (6), (8), and (10) all 1? If so, then SFN is just quantifying the activation after observing the output of the ANN.

5. **Question on the scope of application of the proposed theory and the relationship between MTN and SFN**: Are the MTN and SFN the same concept? Specifically:
   - Theorems 1, 2, and 3 are all proofs for the MTN, yet no proof is provided to verify the relationship between the MTN and the SFN.
   - Why can Theorems 1, 2, and 3 be applied to the SFN and still hold true?

6. **Comment on experimental results**: The experimental results on COCO-2017 object detection may lack persuasiveness, as the backbone networks used (in this work and comparative studies) are not consistent.

**Questions:**

The problem is detailed in the above weaknesses.

---

> ### Author Response · Authors · 2025-11-24
> **Reply to Reviewer pZCi**
>
> 1. **Correctness of Theorem 1**:
>
> In ANN2SNN methods, the threshold is typically set to the maximum or top-p% of the activations (i.e., the largest p% activation value, we set p=1 like [1], [2]), which ensures that almost all inputs are bounded by θ.
>
> 2. **Clarification on the definition of $ x_M $ and $ o_M $**:
>
> We majorly analyze MTN only at a single time step, thus we omit the temporal dimension that appears in the notation for the IF neuron, i.e., we write $ x_M = x_M(1) $, $ o_M = o_M(1) $.
>
> We clarify the notation used in the formulas as follows. In Equation (6), $ o(t) $ denotes the output of MTN over multiple time steps. In Equation (7), $ \bar{o}_{IF} $ denotes the time-averaged output of the IF neuron over multiple time steps, and $ o_M $ denotes the single-step output of MTN. In Equation (17), we write $ x_M = x_M(1) $ to indicate the input to MTN at a single time step. In Equation (27), we write $ o_M = o_M(1) $ to indicate the output of MTN at a single time step.
>
> The output equivalence is exactly as described in Equation (32): it means that the average output of the IF neuron in layer $ l $ over T time steps is equal to the output of MTN at the first timestep.
>
> 3. **Necessity of our work**:
>
> Our goal is to propose an ANN2SNN method that not only performs well for T>1, like existing approaches, but also maintains strong performance at T=1, thereby reducing inference latency. To the best of our knowledge, there has been no prior ANN2SNN work that remains non-degenerate at T=1. The differences between such single-step SNNs and quantized ANNs are explained in detail in our common comment.
>
> 4. **Relationship between MTN and SFN**:
>
> SFN is a generalization of MTN, specifically in that:
> - Allowing sparse spike emission.
> - Independently optimized scaling factor.
>
> The motivation for generalizing MTN is discussed in Section 3.1.4 (Limitations and improvements).
>
> Theorems 1 and 2 focus on the equivalence between MTN and IF under idealized conditions, which does not hold for SFN.
>
> Theorem 3 gives a theoretical error bound between MTN and IF under practical conditions. However, the configuration of SFN depends on the specific model/task combination, so Theorem 3 does not directly apply to SFN. The ablation results in Table 5 show that SFN performs better than MTN in practice (in Table 5, the combination Scaling Strategy + linear corresponds to MTN). Therefore, we may regard the error of a properly configured SFN as smaller than that of MTN, and still effectively controlled by the theoretical upper bound in Theorem 3.
>
> 5. **Inconsistent backbone networks**:
>
> Among the ANN2SNN methods we surveyed, we did not find any experimental results on COCO-2017 that use the same backbone networks as ours.
>
> To further support the validity of our approach, we have added a supplementary experiment using YOLOv5s for comparison with SUHD [3] in Table 3.
>
> **References**:
>
> [1] Rueckauer, et al. “Theory and tools for the conversion of analog to spiking convolutional neural networks.”
>
> [2] Huang, et al. “Towards High-Performance Spiking Transformers from ANN to SNN Conversion.”
>
> [3] Qu, et al. "Spiking neural network for ultralow-latency and high-accurate object detection."

---

> ### Author Response · Authors · 2025-11-28
>
> We would like to sincerely thank you again for your constructive comments. In the revised manuscript and rebuttal, we have carefully addressed each of the raised points. We hope that these clarifications and additional results help to resolve your main concerns, and we would very much appreciate it if you could let us know whether the current revision adequately addresses your previous comments.

---

### Author Response · Authors · 2025-11-24
**Common Reply**

We sincerely thank all reviewers for their careful reading of our submission and for the constructive feedback. Several comments from different reviewers converge on a few core aspects of the paper: (i) the comparison between our single-step SNNs and quantized ANNs, (ii) the robustness and cost of our hyperparameters, (iii) the overview contribution of SFN. We summarize our unified responses here and then address reviewer-specific points in the individual replies.

1. **Differences between single-step SNNs with MTN/SFN and quantized ANNs**:
- **Temporal extension** : This paper focuses on T = 1 SNN inference to reduce latency while maintaining accuracy. However, MTN/SFN can be naturally extended to T > 1, with nonlinear modules handled by ECM [1] and membrane potentials updated over time. In this regime, MTN/SFN is quite different to a simple activation quantizer. Table 4 shows that increasing T significantly improves accuracy (e.g., ViT-Base: 70.6% → 77.6%).

- **Compatibility with quantized ANNs**: ANN2SNN methods can be combined with weight quantization, yielding SNNs with quantized weights by treating a weight-only quantized ANN as the “original ANN” for conversion.

- **Energy efficiency of single-step SNNs vs. quantized ANNs**: As an illustrative example, assume all weights are INT8 and consider one linear layer:
  - Single-step SNN with MTN/SFN: MTN/SFN first emits integer spikes via $ O(N) $ (N the number of thresholds) pointwise comparisons. These spikes are then multiplied with INT8 weights, where MAC can be implemented as repeated AC operations in hardware (a standard practice in neuromorphic designs; see Appendix E.3). The resulting integer output is finally scaled by a fixed threshold using fixed-point arithmetic. The dominant cost is AC, with each INT8 AC consuming about 0.03 pJ, while the costs of comparisons and fixed-point scaling are comparatively negligible.
  - Quantized ANN: In a quantized ANN running on general-purpose GPUs, INT8 MAC operations dominate the cost, and additional data-type conversions introduce extra energy consumption.

  Overall, compared with the $ O(Ld_{in}d_{out}) $ ($ L $ the sequence length, $ d_{in}, d_{out} $ the input/output dimensions) AC/MAC cost of the linear layer, the additional pointwise operations in the SNN path contribute only a small fraction of the total energy, while avoiding extra energy consumption from data type conversions. In SNNs, most neurons fire relatively small spike counts (Figure. 4), so the actual number of repeated AC operations remains limited. Using the FP32 energy figures (0.9/4.6) as in Table 4, we already obtain a favorable energy ratio; under the INT8 setting (0.03/0.23), this ratio would be even smaller, leading to additional savings [2].

2. **Hyperparameter computation, sensitivity, and generalization**:
- Base threshold: We set the base threshold as the top-p% activations for each layer using one evaluation pass on the original ANN. A choice of p = 1 generalizes well across models and tasks.
- Number of positive/negative thresholds: Symmetric positive/negative branches are standard in prior work, and the hyperparameter N is shared across model–task pairs. We will add comparisons over different N.
- Firing function: We use a “linear near zero, exponential in the tails” form, which generalizes well across architectures and datasets.
- Softmax-specific thresholds: These are estimated in the same pass as (a) and, as Table 7 shows, are very similar across model–task pairs.
- Scaling factor: After (a), we optimize the scaling factor independently, the cost is reported in Appendix E.2, and once this optimization is done, the resulting value can be reused for all subsequent runs of the same model–task setting. Table 7 shows that it falls in a similar range across settings, and Fig. 3(a) indicates that performance is stable if it stays in a reasonable interval. If one does not tune the scaling factor, a coarse choice of λ=0.3 is also robust.

In summary, these hyperparameters have been carefully tuned and show 	consistent generalization across different models and tasks, ensuring stable performance and robustness in practical applications.

3. **The overview contribution of SFN**:

We will include it in the supplement to the common reply.

**References**:

[1] Huang, et al. “Towards High-Performance Spiking Transformers from ANN to SNN Conversion.”

[2] Neseem, et al. “PikeLPN: Mitigating Overlooked Inefficiencies of Low-Precision Neural Networks.”

---

### Author Response · Authors · 2025-11-24
**Supplement to the Common Reply**

3. **The overview contribution of SFN**:

Our main contribution of SFN extends MTN-style designs in two key ways. First, we explicitly introduce a scaling factor whose role is theoretically characterized by Temporal-to-Spatial Equivalence: it should be proportional to the minimal non-degenerate time steps T required by the model. Unlike prior MTN methods that lack such a factor and consequently degrade severely at T=1, our ablations in Table 5 show that the scaling factor is crucial for maintaining high single-step accuracy. Second, SFN employs a sparse firing mechanism whose thresholds are adapted to the activation distribution, in contrast to fixed, exponentially spaced thresholds in [1]; this distribution-aware design further improves accuracy, as evidenced by the comparisons of different firing schemes in Table 5. Taken together, these components yield an ANN2SNN framework that can naturally extend to T>1 with genuine spiking dynamics while remaining non-degenerate at T=1. To the best of our knowledge, this is the first ANN2SNN method to simultaneously support both low-latency single-step inference and temporally extended spiking behavior. Our experiments in Tables 2 and 3 also demonstrate that, under the same backbone architectures, SFN-based SNNs consistently achieve higher accuracy than existing ANN2SNN methods, and that our approach scales effectively to larger models such as EVA.

**References**:

[1] Huang, et al. “Towards High-Performance Spiking Transformers from ANN to SNN Conversion.”

---

### Comment · Area_Chair_mp9v · 2025-11-24
**Please respond to the authors' rebuttal**

Dear reviewers,

The authors have now posted their rebuttal. Please review it and submit your responses as soon as possible so that they still have adequate time to address any remaining questions or concerns. Please note that the discussion period between authors and reviewers will close on December 3, 11:59 PM AOE, after which no further comments can be exchanged.

@Reviewer aVto: Thank you for already starting the discussion.

Best, Your AC

---

### Author Response · Authors · 2025-12-03

Our ANN2SNN framework primarily makes an algorithmic contribution, while also having the potential to improve energy efficiency on hardware. Regarding the reviewer’s request for a more fine-grained energy analysis, we unfortunately cannot provide measurements on real hardware at this stage. Nevertheless, we have refined our energy model to offer a more comprehensive theoretical analysis, explicitly incorporating additional control overhead and more detailed memory accesses into the estimation.

We consider a standard SFN followed by a linear layer. We decompose the total energy consumption into three parts:

$$
E_{total} = E_{DRAM} + E_{on-chip} + E_{comp}
$$

where $E_{DRAM}$ is the DRAM energy, $E_{on-chip}$ denotes on-chip memory energy, and $E_{comp}$ is the computation energy.

We assume the input has shape $[B, L, d_{in}]$, the output has shape $[B, L, d_{out}]$, and $N$ thresholds for SFN. We assume the average firing rate is $p$.

We adopt the following per-operation energy parameters for a given CMOS technology:

- $E_{DRAM}$: energy of a single DRAM access.
- $E_{SRAM}$: energy of a single on-chip SRAM access.
- $E_{REG}$: energy of a single register-file access.
- Computation operation energies such as comparison, addition, multiplication.

### DRAM energy

The weights of the linear layer and the $N$ thresholds are first read from DRAM and then kept on-chip. Since $N \ll d_{in}d_{out}$, the DRAM energy of the SNN is essentially the same as that of the corresponding ANN.

### On-chip memory energy: SRAM + Register

In a hardware implementation, each word read from SRAM is first loaded into the register file, and write-back usually also passes through registers. A simple idea is to assume that each SRAM access is accompanied by one register-file access. In that case, the on-chip memory energy of the SNN becomes

$$
E_{on-chip}^{SNN} = N_{SRAM}^{SNN}(E_{SRAM} + E_{REG})
$$

More concretely, we can decompose $N_{SRAM}^{SNN}$ into several parts:

$$
N_{SRAM}^{SNN} = N_{SRAM}^{V-mem} + N_{SRAM}^{th} + N_{SRAM}^{fire} + N_{SRAM}^{AC} + N_{SRAM}^{scale}
$$

where

- $N_{SRAM}^{V-mem}=3BLd_{in}$ counts accesses for initiately updating $V=V+x$.
- $N_{SRAM}^{th}=NBLd_{in}$ counts accesses for $N$-step threshold comparison.
- $N_{SRAM}^{fire} \leq 4NBLd_{in}$ counts accesses for $N$-step membrane updating and spike outputs.
- $N_{SRAM}^{AC}=3pBLd_{in}d_{out}$ counts accesses during the AC operations of the linear layer.
- $N_{SRAM}^{scale}=2BLd_{out}$ counts accesses for the fixed-point scaling by the base threshold.

The total on-chip memory energy of SNN is

$$
E_{on-chip}^{SNN} \leq (3+5N+3pd_{out})BLd_{in}(E_{SRAM} + E_{REG}) + 2BLd_{out}(E_{SRAM} + E_{REG})
$$

Comparing to the total on-chip memory energy of ANN

$$
E_{on-chip}^{ANN} = (3BLd_{in}d_{out}+BLd_{out})(E_{SRAM} + E_{REG})
$$

We have $ E_{on-chip}^{SNN} \approx E_{on-chip}^{ANN}$ since $N \ll pd_{out}$, the dominant component of each is same.

### Computation energy

As shown in the previous section, we can decompose $E_{comp}^{SNN}$ into several parts:

$$
E_{comp}^{SNN} = E_{comp}^{V-mem} + E_{comp}^{th} + E_{comp}^{fire} + E_{comp}^{AC} + E_{comp}^{scale}
$$

where

- $E_{comp}^{V-mem}=BLd_{in}E_{add}$ counts for initiately updating $V=V+x$.
- $E_{comp}^{th}=NBLd_{in}E_{compare}$ counts for $N$-step threshold comparison.
- $E_{comp}^{fire} \leq 2NBLd_{in}E_{add}$ counts for $N$-step membrane updating and spike outputs.
- $E_{comp}^{AC}=pBLd_{in}d_{out}E_{add}$ counts for AC operations of the linear layer.
- $E_{comp}^{scale}=BLd_{out}E_{mul,fix}$ counts for the fixed-point scaling by the base threshold.

The total computation energy of SNN is

$$
E_{comp}^{SNN} \leq (1+2N+pd_{out})BLd_{in}E_{add} + NBLd_{in}E_{compare} + BLd_{out}E_{mul,fix}
$$

Comparing to the total computation energy of ANN

$$
E_{comp}^{ANN} = BLd_{in}d_{out}E_{mul}
$$

We have

$$
\frac{E_{on-chip}^{SNN}}{E_{on-chip}^{ANN}} \approx \frac{pE_{add}}{E_{mul}}
$$

since $N \ll pd_{out}$, and the energy ratio of this dominant component is the same as our analysis in our paper.

### Summary

Under this more fine-grained energy model, we expect most of the energy savings to come from the computation term rather than from memory accesses. In our ANN2SNN setting, the average firing rate $p$ of the SNN needs to match the ANN activations, which makes $p \approx 1$ and therefore leads to similar memory-access energy between ANN and SNN. By contrast, the additional control overhead introduced in the SNN (e.g., threshold comparisons and membrane updates) accounts for only a small fraction of the total computation energy, so the dominant energy ratio is still governed by the AC vs. multiply operations and remains roughly consistent with the values reported in Table 4 of our paper.

Finally, we sincerely appreciate all reviewers’ insightful comments and suggestions, and we will consider concrete hardware implementations as an important direction of our future work.

---

### Meta-Review · Area_Chair_Ysz8 · 2025-12-19

**Summary:**

This paper proposes a novel method (SFN) for ANN-to-SNN conversion, aiming to achieve high-performance inference at a single timestep (T=1), supported by a "Temporal-to-Spatial Equivalence" theory. The reviewers' core concerns revolve around four main areas: 1) The Nature and Necessity of the Work: Several reviewers (pZCi, aVto, 37fc) question whether the method degenerates to simple activation quantization at T=1, the validity of its "spiking" characteristics and energy advantages, and whether it constitutes a genuine ANN-SNN conversion. 2) Theoretical and Methodological Rigor: Concerns were raised about the premise of Theorem 1 (bounded input) and the necessity of the theoretical contribution (especially Theorem 2) (pZCi); the relationship between the theoretical model (MTN) and the proposed model (SFN); and the handling of negative activations and nonlinear components like LayerNorm (pZCi, Q445). 3) Sufficiency of Empirical Validation: This includes the fairness of comparisons with other ANN2SNN and quantization methods, the completeness of the energy estimation model (whether it accounts for memory access and threshold comparison overheads), and the lack of hardware implementation or simulation evidence (37fc, aVto, Q445). 4) Hyperparameter Sensitivity and Generalization: Concerns exist regarding the robustness and generalizability of the numerous calibrated hyperparameters in SFN (e.g., scaling factor λ, thresholds) across different tasks/models (aVto, Q445).

**Reviewer Concerns:**

The addressed/partially clarified concerns:

1.pZCi (Theorem 1 condition, Notation): Authors clarified how thresholds are set in ANN2SNN to bound inputs and explained the notation for outputs.

2.pZCi & 37fc (Experimental Fairness): Authors added comparisons with ECMT (same ViT architecture), reported results on VGG-16 with relative accuracy drop, and added a YOLOv5s comparison for object detection.

3.37fc & aVto (Comparison to Quantization): Authors added a detailed comparison table with various PTQ methods, acknowledging their SNN may trail in accuracy at T=1 but arguing for energy efficiency.

4.Q445 (Nonlinearity Handling, N-sweep, Search Cost): Authors explained using ECM for T>1 to handle nonlinearities, added an N-sweep experiment (Table 7), and described the search cost for λ in the appendix.

5.Q445 (Originality): Authors differentiated SFN by emphasizing the explicit, optimized scaling factor and flexible firing scheme.

The concerns that are still outstanding (Core Issues):

1.The Fundamental Debate on Contribution (pZCi, aVto): The rebuttal did not resolve the central skepticism that a T=1 "SNN" is functionally a quantized ANN. Reviewers' questions about the necessity of Theorem 2 and whether this work represents a meaningful advance in spiking neural networks remain potent.

2.Empirical Proof for Energy Claims and Hardware Feasibility (37fc, aVto): This is the most critical unresolved cluster. Authors rely on an op-count energy model (0.9pJ/AC) which reviewers explicitly reject as inapplicable to SFN's non-binary operations. The authors provided no hardware simulation, implementation strategy, or energy breakdown to counter these demands. The claims of energy efficiency and asynchronous execution remain unsubstantiated by the evidence required by the reviewers.

3.The concern proposed by Reviewers  (aVto, Q445)  of method Complexity and Generalization is not addressed. While authors provided some stability analysis, concerns about the method's reliance on multiple calibrated heuristics and its generalizability "out-of-the-box" to new models/tasks persist.

**Reviewer Scores:**

Reviewer pZCi (Initial: 4, "marginally below acceptance"): The author's clarifications on theory and notation were likely appreciated. However, the core question about the work's essence (ANN quantization vs. SNN conversion) remains. It is unlikely the reviewer's fundamental view shifted significantly. Probable Final Score: 4.

Reviewer 37fc (Initial: 2, "reject"): The reviewer acknowledged some improved comparisons but doubled down on the critical flaw of the energy model in their rebuttal response. Since the authors did not address the specifics of memory access costs and multi-threshold overhead, the reviewer's reason for rejection stands. Probable Final Score: 2 (Reject).

Reviewer aVto (Initial: 2, "reject"): The reviewer's concerns are the deepest and most comprehensive. The authors' responses, including asking which open-source tools to use, likely reinforced the reviewer's view that the hardware claims are unsupported. None of the major criticisms were convincingly refuted with new evidence. Probable Final Score: 2 (Reject).

Reviewer Q445 (Initial: 4, "marginally below acceptance"): This reviewer was the most constructive and received detailed, point-by-point responses. This could have improved their perception of the paper's rigor. In isolation, they might have considered raising their score. However, in a full discussion, the severe and unresolved issues highlighted by Reviewers 37fc and aVto—particularly regarding the unvalidated energy model—would heavily influence a balanced assessment. Probable Final Score: 4 (likely unchanged due to the weight of outstanding critical concerns from other reviewers).

---

### Decision · Program_Chairs · 2026-01-26

Reject